# Benford's Law as a Distributional Prior for Post-Training Quantization of Large Language Models

## Abstract

Post-training quantization (PTQ) is a practical way to reduce the memory footprint of large language models, but low-bit quantization is sensitive to mismatches between the quantization codebook and the empirical weight/activation distributions. We revisit Benford-like leading-digit statistics as a lightweight diagnostic of scale-broad behavior in transformer tensors. Across several model families, we observe a consistent functional dichotomy: transformational `nn.Linear` weights tend to be Benford-like, whereas `LayerNorm` parameters systematically deviate. Motivated by this observation, we propose BENQ, a data-free PTQ codebook that uses a simple log-spaced grid as a proxy for scale-broad distributions and applies it selectively to transformational layers while keeping stability-critical parameters in higher precision. In 4-bit group-wise PTQ, BenQ consistently improves over uniform RTN and trades wins with NF4 across architectures and tasks, while remaining substantially simpler than optimization-based methods. We additionally report dynamic activation quantization as an exploratory stress test: the results show that log-spaced grids can reduce RTN failures in some families, but also reveal that outlier handling remains essential for reliable low-bit activation PTQ.

## 1 Introduction

Large Language Models (LLMs) deliver state-of-the-art results across NLP tasks, yet their memory and latency footprints hinder broad deployment (Touvron et al., 2023; Jiang et al., 2023). Post-training quantization (PTQ) is a practical remedy: by mapping full-precision weights to few-bit integers, it compresses models and often accelerates inference with modest accuracy loss (Frantar et al., 2022; Dettmers et al., 2023). The de-facto baseline, Round-To-Nearest (RTN) on a *uniform* grid, is simple and hardware-friendly, but it implicitly assumes that parameters occupy the dynamic range evenly.

Empirically, neural weights are highly non-uniform and concentrate near zero (Han et al., 2015). In low-bit regimes (e.g., 3-4 bits), uniform grids spend disproportionate capacity on rare large magnitudes while under-resolving dense near-zero regions; the mismatch is exacerbated in layers whose weight magnitudes span multiple decades. This has spurred a broad literature on *non-uniform* or *distribution-aware* quantization, from classic logarithmic level schedules (Miyashita et al., 2016) to modern per-layer, learned, or optimized codebooks for LLMs (Zhao & Yuan, 2025). In parallel, activation-aware schemes such as AWQ (Lin et al., 2024) and SmoothQuant (Lin et al., 2024) reduce sensitivity to outliers and can be combined with weight-only PTQ (Lin et al., 2024; Xiao et al., 2023).

We revisit the *Benford's Law* (BL) (Benford, 1938), a classic regularity of natural data, and show that many *transformational* layers in modern transformers (linear/attention/FFN) exhibit *Benford-like* leading-digit statistics, whereas normalization layers systematically do not, as illustrated in Figure 3. Beyond empirical evidence, we give a log-domain rationale: multiplicative stochastic optimization (SGD with decay and adaptive preconditioning) induces broad mixtures in $\log|w|$, yielding near-uniform mantissas and thus Benford-like behavior. Notably, prior work has leveraged Benford's Law as an *analysis or training signal*: Sahu et al. (2021b) propose a model as a predictor of generalization and a validation-free early-stopping

criterion, while Ott et al. (2025) regularize significant-digit histograms to improve generalization in low-data regimes.

In this work, we propose **Benford-Quant (BenQ)**, a *data-free* non-uniform PTQ codebook that replaces the linear codebook with a *log-spaced* grid *motivated* by Benford-like scale-broad statistics, and applies it *selectively* to transformational layers while leaving stability-critical parameters (e.g., LayerNorm scales, embeddings) in higher precision. We emphasize that the log grid is a simple *proxy* for scale-broad distributions, rather than a claim that it is an optimal quantizer for a specific analytic prior.

Our contributions are:

- **A diagnostic.** We measure leading-digit (Benford-like) statistics across multiple LLM families and identify a functional dichotomy: transformational `nn.Linear` weights are often Benford-like, while `nn.LayerNorm` parameters systematically deviate.

- **A simple, data-free codebook.** We introduce BenQ, a log-spaced PTQ codebook motivated by scale-broad behavior, designed as a lightweight drop-in replacement for uniform grids in group-wise PTQ.

- **Selective application.** We propose and evaluate a digit-statistics–informed selective strategy that preserves stability-critical parameters in higher precision.

- **Empirical study.** We evaluate 4-bit group-wise PTQ across several model families on perplexity and downstream benchmarks, and report dynamic activation quantization results.

BenQ is a *data-free* codebook intended for lightweight PTQ settings. It is complementary to activation-aware or calibration-heavy methods such as AWQ (Lin et al., 2024), and SmoothQuant (Xiao et al., 2023), and to optimization-based PTQ approaches like GPTQ (Frantar et al., 2022) and SINQ (Müller et al., 2025). Accordingly, our comparisons to AWQ/GPTQ/SINQ are meant to *situate* BenQ relative to stronger, calibration/optimization-based baselines.

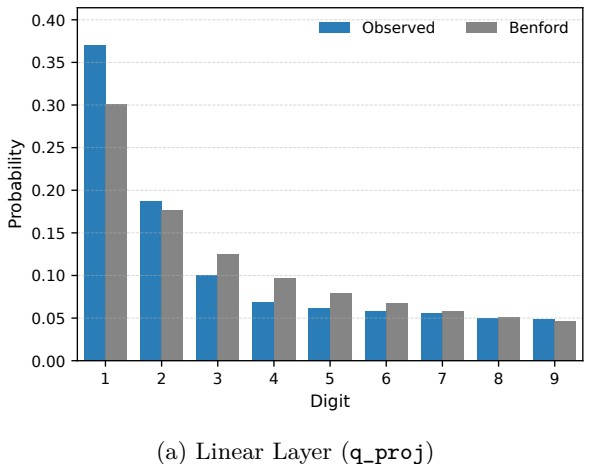

(a) Linear Layer (`q_proj`)

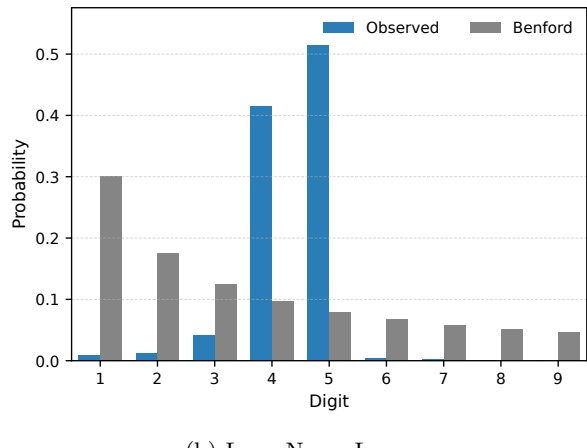

(b) LayerNorm Layer

Figure 1: The dichotomy of Benford's Law compliance in Llama-3-8B. For each digit the right bar indicates Benford's expected probability, while the left bar indicates the observed one. (a) The weights of a transformational linear layer strongly adhere to the Benford distribution. (b) In contrast, the weights of a `LayerNorm` layer systematically violate the law, with their first digits overwhelmingly concentrated on a single value. This analysis motivates our selective strategy (quantize `nn.Linear`, keep `LayerNorm` in higher precision) and supports using a log-spaced codebook as a simple proxy for scale-broad weight distributions.

## 2 Background and Related Work

**Post-training quantization for LLMs.** Post-training quantization (PTQ) compresses pretrained models by mapping full-precision weights to low-bit representations while attempting to preserve accuracy. For LLMs, practical PTQ pipelines commonly rely on group-wise scaling and simple round-to-nearest baselines (RTN), often combined with selective exclusions for stability-critical tensors (Dettmers et al., 2023).

**Non-uniform codebooks.** A long line of work studies non-uniform quantization levels to better match the empirical distribution of neural parameters, including logarithmic schedules (Miyashita et al., 2016; Lee et al., 2017) and more recent approaches that either optimize codebooks or adapt them to data/model statistics (Oh et al., 2021; Wu et al., 2024; Oh et al., 2022). In the LLM literature, NF4 is a prominent non-uniform 4-bit codebook motivated by a normal-quantile construction and widely used in efficient training/serving pipelines (Dettmers et al., 2023). Importantly, these methods differ in their assumptions and operational goals. Some aim for a simple static grid, while others incur optimization or calibration costs to reduce reconstruction error.

**Optimization- and activation-aware PTQ.** Stronger PTQ baselines explicitly optimize quantization error or incorporate activation information. GPTQ uses second-order information to minimize reconstruction loss during quantization (Frantar et al., 2022), while AWQ applies activation-aware scaling to reduce the impact of quantization on salient channels (Lin et al., 2024). Related activation-smoothing approaches such as SmoothQuant similarly aim to mitigate activation outliers (Xiao et al., 2023). SINQ occupies a middle ground: it is calibration-free, yet approximates activation-awareness by recovering column-wise scales from the weight matrix structure via a Sinkhorn-Knopp-style algorithm (Müller et al., 2025). Because these methods are not data-free in the same sense as static codebooks, we treat comparisons to GPTQ, AWQ, and SINQ as context for where a lightweight static codebook stands relative to stronger optimization-, activation-, or scaling-based PTQ methods.

**Benford-like statistics in neural networks.** Benford's Law has been used as an analysis signal in machine learning, including as a diagnostic correlated with generalization and as a validation-free early-stopping criterion (Sahu et al., 2021b), as well as as an explicit regularizer on significant-digit histograms (Ott et al., 2025). Our work differs in goal and scope, rather than using Benford-like statistics as a training signal, we use them as a lightweight diagnostic to motivate a simple log-spaced PTQ codebook and a selective quantization strategy.

### 2.1 Why to Expect Benford Adherence

**Benford preliminaries.** Let $S_{10}(x) \in [1, 10)$ be the base-10 significand, $x = S_{10}(x) \cdot 10^k$. Benford's Law gives

$$\mathbb{P}(\lfloor S_{10}(x) \rfloor = d) = \log_{10}\left(1 + \tfrac{1}{d}\right), \quad d = 1, \ldots, 9. \tag{1}$$

A standard characterization states: $x$ is Benford $\Leftrightarrow$ the fractional part of $\log_{10}|x|$ is uniform on $[0, 1)$ (Hill, 1995b). Thus, Benford-like behavior reduces to *equidistribution modulo* 1 in the log domain.

**Multiplicative training dynamics.** For a scalar weight $w_t$ in a linear/affine map, a broad class of optimizers admits

$$w_{t+1} = (1 - \eta_t \lambda)w_t - \eta_t \phi_t g_t = M_t w_t + \varepsilon_t, \tag{2}$$

where $\lambda \geq 0$ (decoupled weight decay), $\eta_t$ is the step size, $\phi_t > 0$ a preconditioner (e.g., Adam), $g_t$ a stochastic gradient, and $\varepsilon_t$ an additive residual. When $|M_t w_t|$ dominates $|\varepsilon_t|$ during nontrivial epochs, the magnitude evolves approximately multiplicatively:

$$\log|w_{t+1}| \approx \log|w_t| + \log|M_t|. \tag{3}$$

Random fluctuations in $\eta_t, \phi_t$ and data/curvature induce a noisy random walk in $\log|w_t|$, producing broad log-distributions (often close to lognormal). Aggregating across layers and training phases yields mixtures of such log-broad components.

**Matrix products and Benford.** Beyond temporal evolution, the *spatial structure* of neural networks also promotes Benford-like behavior. Each forward pass involves repeated matrix–vector multiplications,

$$\mathbf{h}_{\ell+1} = \mathbf{W}_\ell \mathbf{h}_\ell, \tag{4}$$

so entries of $\mathbf{h}_{\ell+1}$ are sums of products of the form $\prod_{j=1}^{k} w_j h_j$. Classical results (e.g., Hill 1995a;c) show that products of independent, non-degenerate random variables tend to produce significands that are uniformly distributed in the log domain, hence Benford. In deep networks, activations at later layers accumulate multiplicative contributions from many random weights, further broadening the log-distribution of effective coefficients. This complementary perspective explains why even static weight matrices (not only their SGD trajectories) naturally exhibit Benford-like first-digit histograms.

**Consequence: near-uniform log mantissas.** Broad, continuous log-distributions and random mixtures, although not an indisputable rule as stated by Berger & Hill (2019; 2011), are known mechanisms that can lead to equidistribution of $\{\log_{10} |w|\}$ modulo 1, hence Benford-like leading digits. This rationale applies to *transformational* weights (linear/attention/FFN). Parameters tightly anchored to a narrow scale—e.g., LayerNorm scales acting as learned damping factors—*violate* the broadness condition and need not be Benford, matching our empirical dichotomy.

**Relation with Thermodynamics.** Beyond the aforementioned motivation, Sahu et al. (2021b) justify the adherence of neural network weights to Benford's Law through an analogy between thermodynamic systems seeking equilibrium and the training process of neural networks. In this scenario, the weights are associated with particle energy states, and gradient descent is related to the system's temperature. As occurs in systems that follow Boltzmann–Gibbs statistics (e.g., an ideal gas in a closed chamber), changes in temperature induce fluctuations in the distribution of the mantissas of particle energy states around Benford's Law.

Still regarding this work, empirical analyses on the matter are also presented. In particular, using the Model Enthalpy (MLH) metric proposed by the authors, significant adherence to Benford's Law was observed across a variety of architectures, such as CNNs, LSTMs, and early pretrained Transformer-based models (e.g., BERT, ELECTRA, RoBERTa).

**Implication for quantization.** If $\{\log_{10} |w|\}$ is near-uniform, mass is spread across decades, with highest density near zero. A *log-spaced* grid allocates more levels where weights concentrate, reducing expected distortion at fixed bit width. Conversely, for narrow-scale parameters (e.g., LayerNorm), log spacing is suboptimal, motivating *selective* application.

Although not explicitly framed within the context of Benford's Law, there are quantization studies in the literature on the generation of logarithmic levels that demonstrate their effectiveness in better representing the distributions of weights and activations, especially in image classification tasks (Oh et al., 2021; Wu et al., 2024). Moreover, logarithmic levels also stand out for their hardware efficiency, as they enable the use of optimized operations for such representations, e.g., bit-shifting (Lee et al., 2017).

*Note.* Benford compliance (uniform mantissa in $\log_{10}$) does not imply a strictly log-uniform density; our grid is a practical proxy that captures the near-zero concentration that matters for quantization.

## 2.2 Baseline Quantization Methods

Post-training quantization maps a full-precision tensor $\mathbf{W} \in \mathbb{R}^{m \times n}$ to low-bit integers $\mathbf{W}_q$ via a quantizer $Q(\cdot)$ and dequantizer $DQ(\cdot)$, minimizing $\|\mathbf{W} - DQ(Q(\mathbf{W}))\|$.

**Uniform RTN.** A $B$-bit symmetric uniform quantizer uses $2^B$ evenly spaced levels with scale $s$:

$$\mathbf{W}_q = \text{clip}\Big(\text{round}(\mathbf{W}/s), -2^{B-1}, 2^{B-1} - 1\Big), \tag{5}$$

$$DQ(\mathbf{W}_q) = \mathbf{W}_q \cdot s. \tag{6}$$

Choosing $s$ by $\max|\mathbf{W}|/(2^{B-1}-1)$ is outlier-sensitive; group-wise scaling mitigates this by partitioning $\mathbf{W}$ and using per-group scales (Dettmers et al., 2023). This method was chosen as a way to compare the efficiency of logarithmic levels against uniform ones.

**NF4 Quantization.** A NF4 quantizer uses non-uniformly spaced levels with scale $s$ as follows:

$$\mathbf{W}_q = \arg\min_{v \in \mathcal{V}_{\text{NF4}}} \left| \frac{\mathbf{W}}{s} - v \right|, \tag{7}$$

$$DQ(\mathbf{W}_q) = \mathcal{V}_{\text{NF4}}[\mathbf{W}_q] \cdot s, \tag{8}$$

where $s = max(|\mathbf{W}|)$ and $\mathcal{V}_{\text{NF4}} = $ [-1, -0.696, -0.525, -0.395, -0.284, -0.185, -0.091, 0, 0.08 , 0.161, 0.246, 0.338, 0.441, 0.563, 0.723, 1] (rounded to three decimal places) Dettmers et al. (2023). This method was selected in order to compare the logarithmic levels of BENQ with other types of non-uniform levels (namely, normal levels).

**State-of-the-art Methods.** We additionally compare BENQ to GPTQ (Frantar et al., 2022), AWQ (Lin et al., 2024) and SINQ (Müller et al., 2025) to *situate* it relative to stronger calibration/optimization-based PTQ baselines. GPTQ explicitly minimizes reconstruction error using second-order information, while AWQ uses activation-aware scaling to reduce quantization error in salient channels. SINQ is calibration-free yet approximates activation-awareness by applying dual-axis (row and column) scaling via a Sinkhorn-Knopp-style algorithm. These methods typically achieve lower perplexity than static, data-free codebooks; our goal is not to claim superiority, but to understand where a simple log-spaced grid can be a competitive, lightweight alternative or a complementary component in hybrid pipelines.

## 3 The Benford-Quant Method

Benford-Quant is a post-training, data-free quantization method designed to align the quantization grid with the empirically observed logarithmic distribution of transformer weights. The method consists of three core components: (1) a distributional prior based on Benford's Law, (2) a group-wise quantization algorithm that maps weights to a non-uniform, log-spaced grid, and (3) a selective application strategy that targets only transformational layers. For transparency and reproducibility purposes, the source code will be made publicly available later.

**Benford's Law as a Distributional Prior.** Benford's Law (Benford, 1938) states that the probability of a number having a first significant digit $d \in \{1, \dots, 9\}$ is given by:

$$P(d) = \log_{10}\left(1 + \frac{1}{d}\right). \tag{9}$$

This distribution arises from processes involving scale invariance and implies that values are distributed logarithmically across orders of magnitude. We use this principle as a motivating signal for scale-broad behavior in certain tensors, using it to inform the geometry of a simple log-spaced quantization grid.

**Log-Uniform Quantization Grid.** Motivated by Benford-like scale-broad statistics (rather than claiming an optimal codebook for Eq. (9)), we construct a non-uniform set of $2^B$ quantization levels, $\mathcal{L}$, that are spaced logarithmically.

For quantization with $B$ bits, we generate $(2^{B-1}-1)$ positive levels, $\mathcal{L}^+$, within the normalized range $(\epsilon, 1]$, and include an explicit zero level. These positive levels are evenly spaced in the log domain, concentrating representational capacity near zero

$$\mathcal{L}^+ = \left\{ \exp\left( \log(\epsilon) + i \cdot \frac{\log(1) - \log(\epsilon)}{(2^{B-1}-1) - 1} \right) \,\middle|\, i = 0, 1, \dots, (2^{B-1}-1) - 1 \right\}, \tag{10}$$

and the negative ones are described as

$$\mathcal{L}^- = \left\{ -\exp\left( \log(\epsilon) + i \cdot \frac{\log(1) - \log(\epsilon)}{(2^{B-1} - 1)} \right) \;\middle|\; i = 0, 1, \ldots, (2^{B-1} - 1) \right\}. \tag{11}$$

The full grid is then:

$$\mathcal{L} = \mathcal{L}^- \cup \{0\} \cup \mathcal{L}^+, \tag{12}$$

yielding exactly $2^B$ levels in $[-1, 1]$. This design inherently allocates more representational capacity to the more frequent low-magnitude weights.

Still on the grid construction process, $\epsilon$ is computed once per model (*i.e.* $\epsilon$ does not vary between executions of the same model) as

$$\epsilon = 10^\omega$$
$$\omega = \left\lfloor \log_{10}\left( \frac{Q_{0.999}(|\mathcal{W}|)}{\max |\mathcal{W}|} \right) \right\rfloor, \tag{13}$$

where $Q_{0.999}$ is the quantile at level 0.999 and $\mathcal{W}$ are the sampled weights from the model. This quantity captures the logarithmic displacement between the bulk of the weight distribution (represented by $Q_{0.999}$) and its absolute peak, yielding a non-positive integer that encodes how many orders of magnitude separate the two. Finally, $\mathcal{W}$ is composed of random samples drawn independently from each `nn.Linear` weight tensor, each capped at 100,000 elements, reducing the computational cost of quantile estimation while preserving a statistically representative picture of the global weight distribution.

We emphasize the importance of this $\epsilon$ selection step: early experiments with arbitrarily chosen values of $\epsilon$ yielded severe perplexity degradations, and even a generally well-suited fixed value (*e.g.*, $\epsilon = 10^{-2}$) failed to generalize across all model families. Equation 13 therefore provides a principled, distribution-aware estimate of $\epsilon$ for BENQ, reducing reliance on arbitrary tuning while improving robustness across model families. Deriving provably optimal or fully adaptive selection strategies remains an interesting direction for future work.

**The Quantization and Dequantization Procedure.** The core procedure applies this non-uniform grid to a weight tensor $\mathbf{W}$ in a group-wise fashion. The full process is detailed in Algorithm 1. For each block of weights $\mathbf{w}_g$ of size $G$, we first compute a scale $s_g = \max(|\mathbf{w}_g|)$ to normalize the block to $[-1, 1]$. Then, each normalized weight is mapped to the index of the closest level in our static grid $\mathcal{L}$.

---

**Algorithm 1** The Benford-Quant Quantization Procedure

---

**Require:** Weight tensor $\mathbf{W}$, bit-width $B$, group size $G$.
**Ensure:** Quantized indices $\mathbf{W}_q$, scales $\mathbf{S}$.
 1: $\mathcal{L} \leftarrow$ GenerateLogUniformLevels($B$)            ▷ Pre-compute the $2^B$ non-uniform levels in $[-1, 1]$
 2: $\mathbf{W}' \leftarrow$ reshape($\mathbf{W}, (-1, G)$)            ▷ Reshape W into blocks of size G
 3: Initialize empty tensors $\mathbf{W}_q$ and $\mathbf{S}$ for outputs.
 4: **for** each block $\mathbf{w}_g$ in $\mathbf{W}'$ **do**
 5:     $s_g \leftarrow \max(|\mathbf{w}_g|)$                          ▷ Compute the block's scale
 6:     $\hat{\mathbf{w}}_g \leftarrow \mathbf{w}_g / s_g$             ▷ Normalize block to $[-1, 1]$
 7:     $\mathbf{i}_g \leftarrow \arg\min_{j \in \{1, .., 2^B\}} |\hat{\mathbf{w}}_g^{\mathrm{unsqueeze}} - \mathcal{L}_j|$     ▷ Find index of nearest level for all values in the block
 8:     Append $\mathbf{i}_g$ to $\mathbf{W}_q$; Append $s_g$ to $\mathbf{S}$
 9: **end for**
10: **return** $\mathbf{W}_q, \mathbf{S}$

---

The dequantization process is a simple reversal. Given the integer indices $\mathbf{W}_q$ and the scales $\mathbf{S}$, the reconstructed weight tensor $\tilde{\mathbf{W}}$ is obtained by first performing a lookup into the level grid and then rescaling each block:

$$\tilde{\mathbf{w}}_g = \mathcal{L}[\mathbf{i}_g] \cdot s_g \tag{14}$$

where $\mathcal{L}[\mathbf{i}_g]$ denotes the element-wise lookup operation for the indices corresponding to block $g$.

**Selective Quantization Strategy.** Our empirical findings in Section 4.1 reveal that `LayerNorm` weights do not follow the logarithmic distribution assumed by our method. Applying a log-uniform quantizer to their tightly clustered, near-constant distributions is theoretically and practically suboptimal. We therefore adopt a selective quantization strategy: only transformational layers are quantized. LayerNorm layers are maintained in native precision, as their parameters do not follow BL and incur negligible memory overhead. Token embedding layers do not exhibit the same systematic BL deviation as LayerNorm, but we keep them in higher precision in the main configuration to avoid degradation at the vocabulary interface.

Notably, our ablation study (see Table 13) reveals that these two choices carry asymmetric importance: excluding normalization layers from quantization yields a far greater quality improvement than excluding embedding layers, suggesting that preserving `LayerNorm` precision is the dominant factor in the selective quantization policy.

Finally, we point out that this policy of excluding `LayerNorm` and `Embedding` layers in quantization is not new in the literature, and it is the default behavior in many PTQ frameworks, such as the Hugging Face Transformer framework (Wolf et al., 2019). However, this exclusion has traditionally been motivated by practical considerations, such as the known sensitivity of normalization layers to precision reduction (Li et al., 2024) and the important role of `Embedding` layers as the model's vocabulary interface (Vaswani et al., 2017), rather than by any analysis of the underlying weight distributions. To the best of our knowledge, no prior work has grounded this policy in Benford's Law adherence.

## 3.1 Group-Adaptive Version

In view of the original goal of the method to avoid wasting precision on values that are rarely or not used, we propose a set of quantization levels adapted to the numerical distribution of each quantized group. In this adaptation, the quantization process analyzes each weight group individually to determine its minimum value $g_{min}$ and maximum value $g_{max}$, and subsequently generates a log-uniform grid centered at zero within this range. Algorithm 2 presents the described procedure in detail.

---

**Algorithm 2** Group-Adaptive Benford-Quant

**Require:** Weight tensor $\mathbf{W}$, bit-width $B$, group size $G$.
**Ensure:** Quantized indices $\mathbf{W}_q$, scales $G_{min}$ and $G_{max}$.
 1: $\mathbf{W}' \leftarrow \text{reshape}(\mathbf{W}, (-1, G))$                        ▷ Reshape W into blocks of size G
 2: Initialize empty tensors $\mathbf{W}_q$ and $G_{min}, G_{max}$ for outputs.
 3: **for** each block $\mathbf{w}_g$ in $\mathbf{W}'$ **do**
 4:      $g_{min} \leftarrow \min(\mathbf{w}_g)$
 5:      $g_{max} \leftarrow \max(\mathbf{w}_g)$
 6:      $\mathcal{L} \leftarrow \text{GenerateLogUniformLevels}(B, g_{min}, g_{max})$          ▷ Log-uniform levels in $[g_{min}, g_{max}]$
 7:      $\mathbf{i}_g \leftarrow \arg\min_{j \in \{1,..,2^B\}} |\hat{\mathbf{w}}_g^{\text{unsqueeze}} - \mathcal{L}_j|$      ▷ Find index of nearest level for all values in the block
 8:      Append $\mathbf{i}_g$ to $\mathbf{W}_q$; Append $g_{min}$ to $G_{min}$; Append $g_{max}$ to $G_{max}$.
 9: **end for**
10: **return** $\mathbf{W}_q, G_{min}, G_{max}$

---

Again, the dequantization process is simply the reverse operation. That is, the tensor $\tilde{\mathbf{W}}$ is reconstructed through lookups in the grids defined by $G_{min}$ and $G_{max}$.

# 4 Experiments and Results

Our experiments are designed to answer our core research questions. We evaluate on several transformer-based model families: Gemma3, Qwen and Qwen3, Llama3, OPT, BLOOM, TinyLlama and DeepSeek-R1 (Team et al., 2024; Bai et al., 2023; Grattafiori et al., 2024; Zhang et al., 2022; Workshop et al., 2022; Zhang et al., 2024; Guo et al., 2025); and on different tasks: Perplexity, LAMBADA (Paperno et al.,

2016), HellaSwag (Zellers et al., 2019) and MMLU (Hendrycks et al., 2020). All perplexity (defined as $e^{H(p)}$, where $H(p)$ is the entropy of the model's prediction) evaluations are conducted on the test split of WikiText-2 (Merity et al., 2016). The other tasks were evaluated through EleutherAI's LLM evaluation framework, namely *lm-eval*. A single computer equipped with an AMD Ryzen Threadripper 7960X 24-Cores @ 5360MHz, 256 GB of DDR5 RAM and a NVIDIA H200 GPU was used for the majority of experiments. The sole exception occurred in the experiments regarding BenQ (GA) computational overhead, which used a Google Colab instance equipped with an NVIDIA A100 GPU and 83.5 GB of RAM.

### 4.1 RQ1: Investigating Benford's Law in Transformers

**Setup.** To establish the foundation for our method, we first analyze the distribution of the first significant digit for every parameter tensor in our test models. We compare the observed distribution against the theoretical Benford distribution using the Mean Absolute Deviation (MAD) metric, defined by Cerqueti & Lupi (2023) as

$$MAD = \frac{1}{9} \sum_{i=1}^{9} |p_i - b_i|, \tag{15}$$

where $p_i$ denotes the probability of digit $i$ empirically observed, and $b_i$ the corresponding Benford-expected probability. Complementarily, we extended the analysis to the joint distribution of the first two significant digits, in which case MAD is calculated as

$$MAD_{\text{2-digit}} = \frac{1}{90} \sum_{i=10}^{99} |p_i - b_i|, \tag{16}$$

where $p_i$ denotes the observed probability of two-digit significand $i$, and $b_i = \log_{10}(1+1/i)$ the corresponding generalized Benford probability, for $i = \{10, 11, \ldots, 99\}$.

Furthermore, to make the Benford analysis more formal, we conducted statistical tests to assess models' conformity to Benford's Law. However, this is not a trivial task, particularly in the context of LLMs, which may contain billions of parameters. Cerqueti & Lupi (2023) show that conventional statistical tests (e.g. Pearson's $\chi^2$ test), in large-sample settings, tend to reject the null hypothesis of Benford conformity even in the presence of small and practically negligible deviations from the law. Figure 2 illustrates this phenomenon for the OPT-1.3B model.

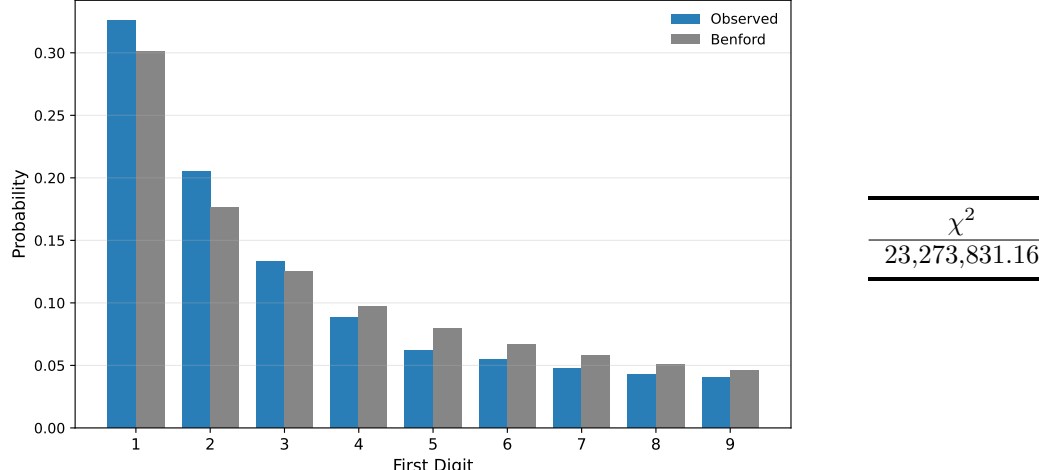

| $\chi^2$ | $p(\chi^2)$ |
|---|---|
| 23,273,831.16 | 0.0000 |

Figure 2: The weights of OPT-1.3B exhibit only small deviations from BL; nevertheless, the $\chi^2$ test rejects the null hypothesis of Benford adherence, highlighting its sensitivity to sample size. This highlights the limitations of conventional statistical tests for assessing Benford conformity on LLMs' weights.

To address this issue, we adopt the $\epsilon$-Benford adherence test proposed by Campanelli (2022), which is asymptotically independent of the sample size. The test is based on a relaxation of the notion of Benford adherence through a tolerance parameter $\epsilon$, whereby a distribution is said to be $\epsilon$-Benford if and only if

$$\left| \frac{P_\epsilon(d) - P_B(d)}{P_B(d)} \right| \leq \epsilon \quad \forall d \in \{1, \ldots, 9\}, \tag{17}$$

where $P_\epsilon(d)$ denotes the observed first-digit probability distribution and $P_B(d)$ the distribution expected under Benford's Law. In other words, the test allows us to assess, with statistical rigor, whether the relative deviation of each digit $d$ with respect to its Benford probability is bounded by a tolerance level $\epsilon$ (null hypothesis) or not (alternative hypothesis). For example, in a 10%-Benford adherent distribution, $P_\epsilon(1) \in [27.09\%, 33.11\%], \ldots, P_\epsilon(9) \in [4.12\%, 5.03\%]$.

For the tests, we adopt a significance level of $\alpha = 0.05$ and set $\epsilon = 0.2$, reporting the resulting test statistics, as well as the minimum $\epsilon$ required for non-rejection of the null hypothesis at the chosen significance level.

**Findings.** Our primary finding is a strong dichotomy based on layer functionality, as illustrated in Figure 3. We consistently observe that weights from transformational `nn.Linear` layers (e.g., in attention and feed-forward blocks) closely follow Benford's distribution. This provides strong motivation for a logarithmically-spaced quantizer. In contrast, `nn.LayerNorm` weights systematically violate the law, with their values clustering around a single learned scalar (e.g., 0.35). We hypothesize these weights function not as transformations, but as learned damping factors to ensure network stability. This key finding motivates our selective quantization strategy, where `LayerNorm` layers are excluded from quantization.

Another finding is the variation in adherence to BL across model families. Figure 3 reveals a sharp split in whether layers comply with BL within the Qwen family. The same does not occur for BLOOM and Gemma3, which exhibit some outliers in their composition, i.e., normalization layers with lower MAD values than other `nn.Linear` layers. Beyond the inherent architectural differences among the models, one hypothesis for this phenomenon is related to the quality of the data used during training and/or the quality of the training process itself, given the correlation between a model's generalization capability and its compliance with BL (Ott et al., 2025; Sahu et al., 2021b;a).

The second-digit analysis yields results consistent with the first-digit findings: `nn.LayerNorm` layers consistently exhibit the highest MAD values across all evaluated models, while transformational layers remain close to the generalized Benford distribution. As expected, the absolute MAD values are smaller in the second-digit analysis, given the larger number of categories (90 pairs vs. 9 digits), but the dichotomy between layer types is preserved. Notably, the `nn.LayerNorm` outliers observed in the Gemma family under the first-digit analysis are substantially reduced in the second-digit analysis, suggesting that the deviations in that family are more concentrated in the first significant digit than in the joint two-digit distribution. Overall, this consistency across both digit orders strengthens the evidence that the Benford-like behavior of transformational layers reflects a broader logarithmic regularity in the weight magnitudes, rather than an artifact of the first-digit distribution alone.

Proceeding to the $\epsilon$-Benford adherence test, Table 1 reports the obtained results. Among the six analyzed models, four are found to be 20%-Benford adherent, with some models, such as Gemma3 1B, exhibiting adherence levels as low as 12.85% (at a significance level of $\alpha = 0.05$). The two models that reject $H_0$, both from the Qwen family, illustrate how normalization layers can negatively affect Benford adherence. As shown in Figure 3, although `nn.LayerNorm` layers are less numerous than `nn.Linear` layers, they are still sufficient to induce rejection of $H_0$. This observation naturally reinforces the motivation for the selective quantization strategy employed in BenQ.

In summary, these experiments provide evidence that: **1.** in general, LLM weights adhere to Benford's Law under a modest $\epsilon$ relaxation; **2.** the required level of relaxation varies across model families, reflecting factors ranging from architectural design to the quality of the training data and procedures; **3.** layer-wise adherence to Benford's Law is closely tied to functional role, with `nn.LayerNorm` layers being considerably less adherent than `nn.Linear` layers; and finally, **4.** our results suggest that model-wise and layer-wise analyses are complementary rather than redundant. Model-wise analysis captures global distributional

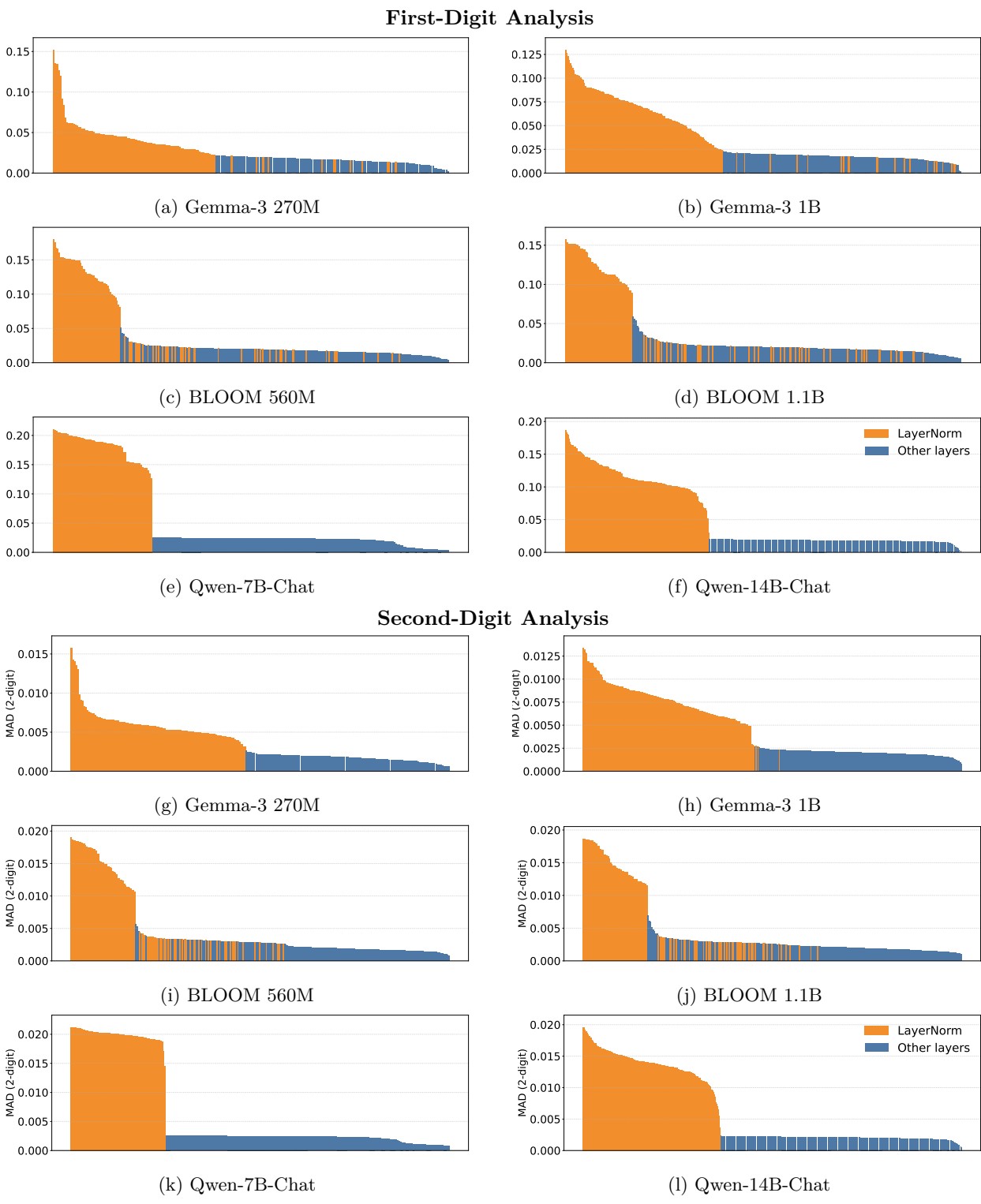

Figure 3: Layer-wise MAD comparison across model families and sizes for first-digit (top, a to f) and second-digit (bottom, g to l) Benford analysis. Orange (hatched) represents normalization layers, while blue represents the other layers (e.g., attention and feed-forward). The dichotomy between transformational and normalization layers is consistent across both digit analyses.

Table 1: $\epsilon$-Benford adherence test results. All model weights are considered in the test. $\epsilon_{\min}$ denotes the minimum value of $\epsilon$ for which the null hypothesis $H_0$ is not rejected at the significance level $\alpha = 0.05$.

| Model | $N$ | $d^*$ | $d_N^*$ | $d_N^*/\sqrt{N}$ | $p(d_N^*)$ | $H_0$ | $\epsilon_{min}$ |
|---|---|---|---|---|---|---|---|
| Gemma-3 270M | 268,098,176 | 0.0540 | 916.6242 | 0.05598 | (0.05, 0.10) | ✓ | 0.1912 |
| Gemma-3 1B | 999,885,888 | 0.0357 | 1169.2854 | 0.03698 | > 0.10 | ✓ | 0.1285 |
| Bloom 1.1B | 1,065,310,419 | 0.0374 | 1263.7890 | 0.03872 | > 0.10 | ✓ | 0.1343 |
| Bloom 560M | 559,213,066 | 0.0436 | 1068.9157 | 0.04520 | > 0.10 | ✓ | 0.1556 |
| Qwen-7B-Chat | 7,721,315,051 | 0.0772 | 7031.6884 | 0.08002 | < 0.001 | | 0.2704 |
| Qwen-14B-Chat | 14,167,276,940 | 0.0725 | 8945.3887 | 0.07515 | < 0.001 | | 0.2544 |

trends and enables cross-model comparisons, but can obscure the heterogeneity that exists across layer types. Conversely, layer-wise analysis exposes this intra-model variation at the cost of a narrower view of the model as a whole. A comprehensive understanding of Benford adherence dynamics in LLMs, therefore, benefits from conducting both analyses simultaneously.

### 4.2 RQ2: Investigating BenQ's Efficiency in Transformers

**Setup.** Having established a potential link to logarithmic distributions in RQ1, we now validate the core design of Benford-Quant. We ask: **1.** can a Benford-inspired grid outperform the traditional uniform one?; **2.** is the Benford-inspired logarithmic spacing of quantization levels essential, or would any non-uniform grid suffice?; **3.** how does Benford-Quant perform in relation to state-of-the-art methods? To answer this, we conducted experiments on several transformer-based models comparing our proposed method against Uniform-RTN, NF4 (Dettmers et al., 2023), GPTQ (Frantar et al., 2022), AWQ (Lin et al., 2024) and SINQ (Müller et al., 2025) baselines. They are detailed in Section 2.2.

It is worth noting that RTN and NF4, like BenQ, were also implemented with selective layer quantization, i.e., keeping normalization and embedding layers unchanged. Therefore, any differences between these methods and BenQ lie in the nature of their quantization grid. Finally, the models quantized with SOTA methods were either downloaded from HuggingFace or quantized with the AutoAWQ/Transformers framework, which also skips quantization of embeddings and normalization layers by default.

**Findings.** Given these considerations, Tables 2, 3, and 4 present the results obtained for the Llama3, Gemma3, and OPT families, respectively, under 4-bit weight quantization with a group size of 128.

Table 2: **4-bit weight-only** group-wise PTQ (group size 128) on the Llama3 family. Best quantized values are shown in bold.

| | Llama3-8.0B model | | | | |
|---|---|---|---|---|---|
| | FP16 | BenQ (GA) | BenQ | NF4 | RTN |
| HellaSwag↑ | 0.606 | 0.590 | 0.590 | **0.592** | 0.590 |
| Lambada↑ | 0.761 | 0.736 | 0.747 | **0.751** | 0.724 |
| MMLU↑ | 0.655 | 0.626 | 0.629 | **0.635** | 0.612 |
| Perplexity↓ | 6.375 | 7.028 | 7.082 | **6.941** | 7.372 |

**Discussion.** Analyzing the questions raised at the beginning of this section, we observe that the Benford-inspired grid frequently outperforms the uniform baseline. For Gemma 3-270M (Table 3), for instance, the log-uniform grid yielded a 10% accuracy improvement on the LAMBADA task compared to the uniform grid. Extending this analysis to the NF4 method, a similar pattern emerges: the normal grid, evaluated on the same model, surpasses the uniform grid by 15%. The Llama3 family (Table 2) also exhibits similar behavior: for Llama3 8B, gains of approximately 2% on the LAMBADA and MMLU tasks were observed

Table 3: **4-bit weight-only** group-wise PTQ (group size 128) on the Gemma3 family. Best quantized values are shown in bold.

| | Gemma3-270M model | | | | | Gemma3-1.0B model | | | | |
|---|---|---|---|---|---|---|---|---|---|---|
| | FP16 | BENQ (GA) | BENQ | NF4 | RTN | FP16 | BENQ (GA) | BENQ | NF4 | RTN |
| HellaSwag↑ | 0.343 | 0.330 | 0.321 | **0.331** | 0.314 | 0.434 | **0.419** | 0.410 | 0.416 | 0.415 |
| Lambada↑ | 0.433 | 0.293 | 0.253 | **0.304** | 0.152 | 0.435 | 0.308 | 0.313 | 0.328 | **0.334** |
| MMLU↑ | 0.268 | 0.245 | **0.272** | 0.268 | 0.237 | 0.396 | **0.367** | 0.339 | 0.347 | 0.350 |
| Perplexity↓ | 24.453 | **36.455** | 44.923 | 40.389 | 57.319 | 28.757 | 41.241 | 41.554 | **36.047** | 44.029 |

Table 4: **4-bit weight-only** group-wise PTQ (group size 128) on the OPT family. Best quantized values are shown in bold.

| | OPT-1.3B model | | | | | OPT-2.7B model | | | | | OPT-6.7B model | | | | |
|---|---|---|---|---|---|---|---|---|---|---|---|---|---|---|---|
| | FP16 | BENQ (GA) | BENQ | NF4 | RTN | FP16 | BENQ (GA) | BENQ | NF4 | RTN | FP16 | BENQ (GA) | BENQ | NF4 | RTN |
| HellaSwag↑ | 0.414 | 0.409 | 0.406 | **0.410** | 0.403 | 0.456 | **0.453** | 0.450 | 0.449 | 0.448 | 0.503 | **0.497** | 0.494 | 0.493 | 0.488 |
| Lambada↑ | 0.588 | 0.573 | **0.581** | 0.577 | 0.567 | 0.643 | 0.621 | 0.612 | **0.625** | 0.617 | 0.677 | 0.671 | 0.665 | **0.676** | 0.661 |
| MMLU↑ | 0.249 | **0.257** | 0.248 | 0.246 | 0.251 | 0.256 | **0.262** | 0.250 | 0.255 | 0.254 | 0.246 | 0.245 | **0.256** | 0.250 | 0.252 |
| Perplexity↓ | 15.125 | **15.753** | 15.882 | 16.000 | 16.180 | 12.807 | 13.556 | 13.597 | **13.381** | 13.860 | 11.114 | 11.372 | 11.445 | **11.294** | 11.814 |

for BENQ over RTN. These results provide evidence that non-uniform levels indeed offer a more suitable representation for LLM weights than uniform grids.

Comparing BENQ against NF4, no consistent superiority is observed in either direction: both methods achieve the best result in different model–metric configurations. across models and metrics, with differences typically within 1-2 percentage points. On Llama3-8B, NF4 outperforms BENQ across all metrics; however, BENQ is favorable in other settings, winning on MMLU across the entire OPT family and on HellaSwag for OPT-2.7B, OPT-6.7B, and Gemma3-1.0B, and achieving lower perplexity than NF4 on OPT-1.3B. These results suggest that, at 4-bit group-wise PTQ, the choice between log-spaced and normal-quantile static grids is architecture- and task-dependent, with neither dominating universally.

We therefore position BENQ not as a replacement for NF4, but as a diagnostic-motivated lightweight alternative: its design is directly grounded in the Benford adherence analysis of Section 4.1, providing an empirically-motivated justification for log-spaced grids in LLM quantization. Like NF4, it requires no calibration data and improves over uniform RTN in most evaluated settings.

**Situating Against SOTA Methods.** Table 5 situates BENQ relative to GPTQ, AWQ, and SINQ, which use calibration/optimization and therefore are expected to achieve lower perplexity. Indeed, these methods consistently improve perplexity in these settings, while BENQ provides a simpler, data-free alternative. This comparison is included for context rather than as a claim that a static codebook should dominate optimized PTQ methods.

Table 5: Quantization results comparing GPTQ, AWQ, and SINQ against BenQ on Llama3-8B and Qwen3-4B (4-bit weight quantization). Best quantized values are shown in bold.

| | Llama3-8.0B model | | | | | Qwen3-4.0B model | | | | |
|---|---|---|---|---|---|---|---|---|---|---|
| | GPTQ | AWQ | SINQ | BENQ (GA) | BENQ | GPTQ | AWQ | SINQ | BENQ (GA) | BENQ |
| Lambada↑ | 0.731 | 0.745 | **0.750** | 0.736 | 0.747 | 0.612 | 0.588 | **0.627** | 0.591 | 0.593 |
| Perplexity↓ | **6.730** | 6.790 | 6.748 | 7.028 | 7.082 | 10.957 | 11.105 | **10.870** | 11.771 | 11.772 |

**Situating Against FP4 Format.** Table 6 compares BENQ with a standard E2M1 FP4 baseline using unquantized FP32 scales. In this idealized quantization-quality comparison, BENQ is competitive with, and often improves upon, the FP4 baseline across the evaluated models and metrics. The largest gains appear

on Gemma3-1B, where BENQ (GA) reduces perplexity from 41.95 to 35.18 relative to FP4. In TinyLlama-1.1B, the advantage is more modest, with BENQ reaching 8.60 perplexity compared with 8.79 for FP4. These results suggest that the Benford-inspired log-spaced grid can provide a useful codebook for weight quantization, while leaving hardware-level efficiency comparisons to optimized FP4 formats as a separate question.

Table 6: Quantization results comparing FP4 (standard E2M1 with scales in FP32) against BENQ on Gemma3-1B, TinyLlama-1.1B-Chat, and Llama3-8B (weight quantization). Group size = 64. Best quantized values are shown in bold.

| | Gemma3-1.0B model | | | TinyLlama-1.1B-Chat model | | | Llama3-8B model | | |
|---|---|---|---|---|---|---|---|---|---|
| | FP4 | BENQ (GA) | BENQ | FP4 | BENQ (GA) | BENQ | FP4 | BENQ (GA) | BENQ |
| HellaSwag↑ | 0.405 | **0.419** | **0.419** | **0.451** | 0.435 | 0.448 | **0.590** | 0.588 | **0.590** |
| Lambada↑ | 0.338 | **0.387** | 0.353 | 0.579 | 0.535 | **0.582** | 0.739 | 0.738 | **0.741** |
| MMLU↑ | 0.356 | **0.375** | 0.350 | 0.250 | **0.262** | 0.255 | 0.579 | 0.629 | **0.634** |
| Perplexity↓ | 41.945 | **35.182** | 36.780 | 8.785 | 9.731 | **8.603** | 7.156 | **6.902** | 6.984 |

**Situating Against Block-Scaling FP Formats.** Tables 7 and 8 situate BENQ relative to NVFP4 (Abecassis et al., 2025) and MXFP4 (Rouhani et al., 2023), respectively. For each, BENQ's group size was adjusted to match the compared method. Under this idealized quantization-quality comparison, BENQ obtains lower perplexity and/or higher downstream scores than the simulated block-FP formats in the evaluated settings. We attribute this outcome to a combination of factors. First, BENQ (GA) employs (adaptive) log-uniform levels per group, fitted to the actual value range of each weight block, whereas NVFP4 and MXFP4 rely on the fixed E2M1 format. Second, BENQ stores scale factors in FP16, while NVFP4 and MXFP4 constrain their scales to 8 bits - E4M3 and E8M0, respectively.

It is important to note, however, that this comparison is not entirely fair: NVFP4 and MXFP4 are formats designed with hardware efficiency as a primary concern, with native support in the Tensor Cores of NVIDIA's Blackwell architecture, which imposes deliberate representational constraints in exchange for substantial gains in throughput and energy efficiency. BENQ, in turn, carries no such hardware commitment, affording it greater freedom in the choice of numerical representations. These results reflect quantization quality under idealized conditions and should not be extrapolated to real-world inference performance, where hardware-native formats such as NVFP4 and MXFP4 hold significant practical advantages.

Finally, comparing NVFP4 against MXFP4, the former achieved better results across all evaluation scenarios. This difference can be explained by three structural factors between the formats. First, NVFP4 uses blocks of 16 elements compared to 32 in MXFP4, providing finer granularity in the scaling process. Second, while MXFP4 stores scale factors in the E8M0 format - restricted to powers of two - NVFP4 employs E4M3 (1 signal bit), which with its 3 mantissa bits allows fractional values and therefore a more accurate approximation of the true distribution within each block. Third, NVFP4 adopts a two-level hierarchical scaling scheme: a local E4M3 scale per 16-element block and a global FP32 scale per tensor, which compensates for the reduced dynamic range of E4M3 relative to E8M0. The results suggest that the gain in fractional precision at the local scale, combined with the finer block granularity, outweighs the dynamic range advantage of E8M0, resulting in lower quantization error and better performance across the evaluated tasks.

Table 7: Quantization results comparing NVFP4 against BENQ on Gemma3-1B, TinyLlama-1.1B-Chat and Llama3-8B (weight quantization). Group size = 16. Best quantized values are shown in bold.

| | Gemma3-1.0B model | | | TinyLlama-1.1B-Chat model | | | Llama3-8B model | | |
|---|---|---|---|---|---|---|---|---|---|
| | NVFP4 | BENQ (GA) | BENQ | NVFP4 | BENQ (GA) | BENQ | NVFP4 | BENQ (GA) | BENQ |
| HellaSwag↑ | 0.399 | **0.423** | 0.420 | 0.439 | **0.459** | 0.452 | 0.581 | **0.597** | 0.594 |
| Lambada↑ | 0.351 | **0.418** | 0.398 | 0.556 | **0.597** | 0.596 | 0.690 | **0.759** | 0.740 |
| MMLU↑ | 0.321 | **0.382** | 0.359 | 0.255 | 0.252 | **0.263** | 0.572 | **0.641** | 0.635 |
| Perplexity↓ | 41.355 | **31.845** | 34.480 | 9.093 | **8.375** | 8.465 | 7.516 | **6.739** | 6.810 |

Table 8: Quantization results comparing MXFP4 against BENQ on Gemma3-1B, TinyLlama-1.1B-Chat and Llama3-8B (weight quantization). Group size = 32. Best quantized values are shown in bold.

| | Gemma3-1.0B model | | | TinyLlama-1.1B-Chat model | | | Llama3-8B model | | |
|---|---|---|---|---|---|---|---|---|---|
| | MXFP4 | BENQ (GA) | BENQ | MXFP4 | BENQ (GA) | BENQ | MXFP4 | BENQ (GA) | BENQ |
| HellaSwag↑ | 0.386 | **0.428** | 0.424 | 0.428 | **0.453** | 0.433 | 0.567 | **0.593** | **0.593** |
| Lambada↑ | 0.301 | **0.429** | 0.392 | 0.521 | **0.602** | 0.526 | 0.681 | **0.751** | 0.741 |
| MMLU↑ | 0.295 | **0.374** | 0.361 | 0.249 | 0.247 | **0.268** | 0.548 | **0.633** | 0.630 |
| Perplexity↓ | 58.146 | **35.527** | 39.134 | 9.864 | **8.446** | 9.773 | 8.063 | **6.824** | 6.891 |

**Bit-Width Sweep.** Tables 9 and 10 present a parameter sweep of BENQ across four bit-widths (2, 3, 4, and 8 bits) on OPT-6.7B and Gemma3-270M models, respectively; and also compare BENQ against RTN on these scenarios. The results indicate that BENQ (especially the GA version) consistently allocates precision more effectively than RTN in low-bit regimes, yielding better performance across most metrics. This advantage is particularly pronounced at 3 bits: on Gemma3-270M, RTN perplexity collapses to 11643.607, whereas BENQ (GA) achieves 460.694—25× lower.

At 8 bits, however, the advantage reverses: RTN achieves lower perplexity on both models, which is consistent with the expectation that uniform quantization suffices when the number of levels (in this case, $2^8 = 256$) is large enough to cover the weight distribution without significant error (Fang et al., 2020). Notably, all methods diverge at 2 bits on both models, suggesting that this regime remains a fundamental challenge regardless of the quantization strategy.

Further on low-bit scenarios, it is also noticeable that BENQ (GA), which generates quantization levels adaptively per group, performs better than BENQ, which relies on fixed log-uniform levels. In this regime, adapting the grid to the local distribution of each group allows BENQ (GA) to allocate levels more precisely where the weight mass is concentrated, while BENQ's fixed grid may misalign with groups whose distributions deviate from the assumed log-uniform prior. As the bit-width increases and more levels become available, this advantage diminishes, which is consistent with the convergence of both variants observed at 8 bits.

Table 9: OPT 6.7B model bit-width sweep.

| | BenQ (GA) | | | | BenQ | | | | RTN | | | |
|---|---|---|---|---|---|---|---|---|---|---|---|---|
| | 2 bits | 3 bits | 4 bits | 8 bits | 2 bits | 3 bits | 4 bits | 8 bits | 2 bits | 3 bits | 4 bits | 8 bits |
| HellaSwag↑ | 0.263 | 0.327 | 0.497 | 0.667 | 0.260 | 0.288 | 0.494 | 0.666 | 0.258 | 0.349 | 0.488 | 0.670 |
| LAMBADA↑ | 0.000 | 0.165 | 0.671 | 0.680 | 0.000 | 0.038 | 0.665 | 0.674 | 0.000 | 0.208 | 0.661 | 0.678 |
| MMLU↑ | 0.229 | 0.244 | 0.245 | 0.245 | 0.249 | 0.235 | 0.256 | 0.249 | 0.229 | 0.250 | 0.252 | 0.247 |
| Perplexity↓ | 8454.311 | 23.664 | 11.372 | 11.220 | 10065.023 | 78.447 | 11.445 | 11.297 | 13574.668 | 58.209 | 11.814 | 11.140 |

Table 10: Gemma3-270M model bit-width sweep. † denotes divergence.

| | BenQ (GA) | | | | BenQ | | | | RTN | | | |
|---|---|---|---|---|---|---|---|---|---|---|---|---|
| | 2 bits | 3 bits | 4 bits | 8 bits | 2 bits | 3 bits | 4 bits | 8 bits | 2 bits | 3 bits | 4 bits | 8 bits |
| HellaSwag↑ | 0.260 | 0.279 | 0.330 | 0.407 | 0.259 | 0.272 | 0.321 | 0.403 | 0.268 | 0.264 | 0.314 | 0.414 |
| LAMBADA↑ | 0.000 | 0.028 | 0.293 | 0.326 | 0.000 | 0.006 | 0.253 | 0.329 | 0.000 | 0.000 | 0.152 | 0.426 |
| MMLU↑ | 0.269 | 0.230 | 0.245 | 0.272 | 0.269 | 0.268 | 0.272 | 0.274 | 0.255 | 0.238 | 0.237 | 0.267 |
| Perplexity↓ | † | 460.694 | 36.455 | 30.756 | † | 1722.632 | 44.923 | 34.328 | † | 11643.607 | 57.319 | 24.363 |

**Group-Size Sweep.** Tables 11 and 12 analyze the effect of the quantization group size on BENQ and BENQ (GA), respectively, using group sizes of 32, 64, 128, and 256. Overall, smaller groups tend to improve quantization fidelity, especially for the Gemma models. For instance, in Gemma3-270M, BENQ reduces perplexity from 51.533 at group size 256 to 38.537 at group size 32, while BENQ (GA) further reduces it from 49.311 to 32.008. A similar pattern is observed in Gemma3-1B, where group sizes 32 or 64 yield the best downstream performance and perplexity. In contrast, OPT-2.7B is less sensitive to group size, with only

modest degradation as the group size increases, suggesting that the interaction between group granularity and codebook shape is architecture-dependent.

These results indicate that fine-grained grouping improves the local alignment between the quantization grid and the empirical weight distribution. Larger groups aggregate more heterogeneous values under a single scale or range, making the effective codebook less well matched to the local distribution and increasing reconstruction error. This effect is particularly visible for BENQ (GA), whose per-group grid benefits from accurately capturing local min–max ranges.

Table 11: BENQ group-size sweep.

| | Gemma3-270M model | | | | Gemma3-1.0B model | | | | OPT 2.7B model | | | |
|---|---|---|---|---|---|---|---|---|---|---|---|---|
| | 32 | 64 | 128 | 256 | 32 | 64 | 128 | 256 | 32 | 64 | 128 | 256 |
| HellaSwag↑ | 0.333 | 0.328 | 0.321 | 0.320 | 0.424 | 0.419 | 0.410 | 0.407 | 0.453 | 0.453 | 0.450 | 0.450 |
| Lambada↑ | 0.292 | 0.244 | 0.253 | 0.223 | 0.392 | 0.353 | 0.313 | 0.292 | 0.623 | 0.619 | 0.612 | 0.610 |
| MMLU↑ | 0.247 | 0.267 | 0.272 | 0.236 | 0.361 | 0.350 | 0.339 | 0.345 | 0.256 | 0.247 | 0.250 | 0.240 |
| Perplexity↓ | 38.537 | 39.880 | 44.923 | 51.533 | 39.134 | 36.780 | 41.554 | 46.979 | 13.515 | 13.533 | 13.597 | 13.745 |

Table 12: BENQ (GA) group-size sweep.

| | Gemma3-270M model | | | | Gemma3-1.0B model | | | | OPT-2.7B model | | | |
|---|---|---|---|---|---|---|---|---|---|---|---|---|
| | 32 | 64 | 128 | 256 | 32 | 64 | 128 | 256 | 32 | 64 | 128 | 256 |
| HellaSwag↑ | 0.331 | 0.327 | 0.330 | 0.324 | 0.428 | 0.419 | 0.419 | 0.408 | 0.454 | 0.453 | 0.453 | 0.451 |
| Lambada↑ | 0.302 | 0.285 | 0.293 | 0.275 | 0.429 | 0.387 | 0.308 | 0.313 | 0.628 | 0.621 | 0.621 | 0.619 |
| MMLU↑ | 0.269 | 0.266 | 0.245 | 0.232 | 0.374 | 0.375 | 0.367 | 0.357 | 0.252 | 0.258 | 0.262 | 0.249 |
| Perplexity↓ | 32.008 | 37.251 | 36.455 | 49.311 | 35.527 | 35.182 | 41.241 | 40.430 | 13.300 | 13.312 | 13.556 | 13.672 |

However, smaller groups also increase metadata overhead. Figure 4 shows the resulting quality–compression trade-off for Gemma3-270M: reducing the group size improves fidelity but decreases the effective compression ratio, with a steeper penalty for BENQ (GA) than for BenQ. This is expected because BENQ (GA) stores additional per-group range information, making it more sensitive to the number of groups. Therefore, group size 32 is preferable when accuracy is prioritized, whereas group size 64 provides a strong practical trade-off between fidelity and compression. We retain group size 128 as a standard setting for comparison with common group-wise PTQ configurations, while the parameter sweep shows that BENQ is not tied to this single operating point.

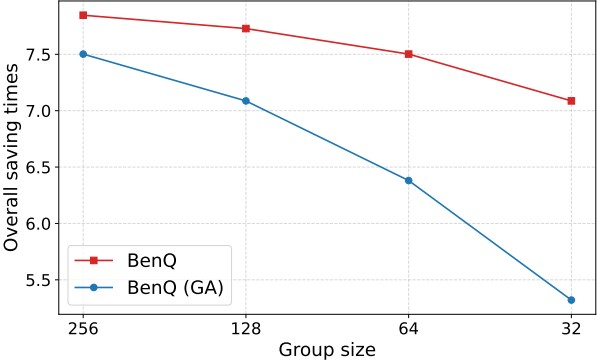

Figure 4: Effective compression ratio for Gemma3-270M, excluding embedding layers, as a function of group size. The steeper curve for BENQ-GA shows that finer granularity incurs higher metadata overhead.

**BenQ's Components Ablation Study.** We ablate the two main design choices of BENQ: the use of logarithmic quantization levels and the selective layer quantization policy. Table 13 reports the resulting

LAMBADA accuracy and perplexity, while Figure 5 shows the perplexity degradation relative to the full configuration.

The ablation shows that excluding normalization layers is substantially more important than excluding embedding layers. With logarithmic levels, excluding only embeddings reduces perplexity only marginally, from 73.90 to 72.41. In contrast, excluding only normalization layers reduces perplexity from 73.90 to 50.07, a drop of more than 23 points. This indicates that quantizing normalization parameters is the main source of instability in this setting, whereas embedding quantization has a comparatively smaller effect for this model.

This result has practical implications for memory-constrained deployment. Embedding layers can account for a large fraction of the total memory footprint in smaller LLMs, as shown in Tables 15 and 16. The component ablation suggests that, when more aggressive compression is required, quantizing embeddings may be a viable option, provided that normalization layers remain in higher precision. We emphasize, however, that this conclusion is based on the evaluated model and should be treated as an empirical indication rather than a universal rule.

Table 13 also reveals that the relative advantage of logarithmic levels depends strongly on which layer types are quantized. When normalization layers are excluded, logarithmic levels outperform uniform ones: in the Norm exclusion setting, perplexity improves from 53.58 with uniform levels to 50.07 with logarithmic levels. However, when normalization layers remain quantized, this advantage disappears. In the Emb exclusion setting, where embeddings are kept in higher precision but normalization layers are still quantized, uniform levels achieve a perplexity of 56.44, whereas logarithmic levels degrade to 72.41.

These results reinforce the Benford-adherence analysis. Normalization parameters deviate substantially from the scale-broad behavior that motivates the log-spaced codebook, making them a poor fit for logarithmic quantization. Conversely, once these non-adherent layers are excluded, the log-uniform grid becomes beneficial for the remaining transformational layers. Thus, the two components of BenQ are complementary, the logarithmic codebook provides the representational advantage, while selective quantization determines where that advantage can be safely expressed.

Table 13: **BenQ's components ablation** for 4-bit weight quantization (group size 128) on DeepSeek-R1-Distill-Qwen-1.5B. Best values across all configurations are shown in bold.

| | Uniform Levels | | | | Logarithmic Levels | | | |
|---|---|---|---|---|---|---|---|---|
| Excluded: | None | Emb. | Norm. | Both | None | Emb. | Norm. | Both |
| Lambada↑ | 0.257 | 0.257 | 0.279 | 0.280 | 0.270 | 0.280 | 0.309 | **0.324** |
| Perplexity↓ | 59.825 | 56.440 | 53.582 | 50.582 | 73.900 | 72.407 | 50.071 | **49.024** |

**BenQ (GA) computational overhead.** As seen in Algorithm 2, BenQ (GA) calculates a different grid for each block $\mathbf{w}_g$. This results in a computational overhead relative to BenQ and NF4, which generate the grid only once. Therefore, concerns about the practical viability of BenQ's adaptation may arise. To address this concern, throughput (tokens per second) was measured during the perplexity evaluation using BenQ (GA), BenQ, and NF4 across group sizes of 128 and 64, as shown in Table 14. The observed gaps are highlighted in Figure 6.

Although BenQ (GA) consistently incurs a throughput penalty across all evaluated models and group sizes, the relative overhead is modest for large models. For BLOOM-3B, OPT-6.7B, and Llama3-8B, the observed degradation relative to BenQ and NF4 ranges from approximately 6% to 8%, depending on the group size. The most pronounced degradation occurs in Gemma-3-270M, where BenQ (GA) reaches losses of up to 35%. However, even in this case, the absolute throughput remains considerably high (above 8,600 tokens/second) in this small-model regime.

The overhead trend is favorable with increasing model size: in the evaluated models, larger architectures exhibit smaller relative slowdowns. This behavior is consistent with the inference cost becoming increasingly dominated by matrix multiplication and feed-forward computation, which dilutes the relative contribution of per-group grid construction.

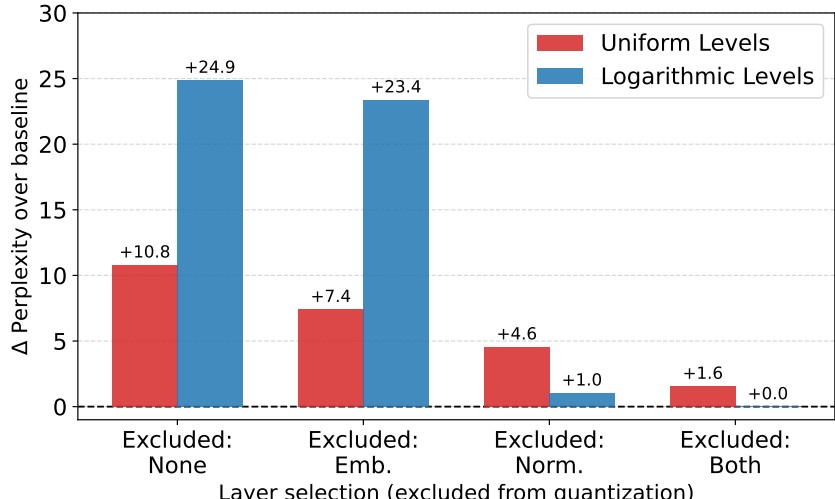

Figure 5: Perplexity degradation under a component ablation study of BenQ (DeepSeek-R1-Distill-Qwen-1.5B, 4-bit, group-size 128). Each configuration reflects the incremental removal of a design choice relative to the full method: logarithmic quantization levels with both normalization and embedding layers excluded from quantization. For example, in the "*Excluded: Norm.*" configuration, only normalization layers are excluded from quantization, while embedding layers remain quantized.

Regarding the effect of group size, the throughput degradation of BenQ (GA) is remarkably stable across $g \in \{64, 128\}$. For all evaluated models, the difference in relative overhead between the two configurations is generally between 1 and 2 percentage points. This is noteworthy because halving the group size doubles the number of quantization groups per layer, which implies twice as many per-group grid calculations. Despite this, the additional cost observed in practice is marginal: doubling the number of per-group grid recalculations does not double the relative throughput degradation.

Finally, the throughput results reported here provide a conservative assessment of BenQ under a standard implementation, as all measurements were obtained without specialized inference kernels. This choice isolates the method's behavior from kernel-level engineering and leaves substantial room for systems optimization. Dedicated kernels, analogous to those developed for GPTQ and AWQ, could reduce the observed overhead in practical deployments, consequently, the reported throughput gaps reflect the current unoptimized implementation rather than an inherent constraint of the method. Kernel-level optimization is an orthogonal extension and is further discussed in Section 5.

Table 14: Average token throughput (tokens/second) by model, method, and group size.

| | Gemma-3-270M | | | BLOOM-3B | | | OPT-6.7B | | | Meta-Llama-3-8B | | |
|---|---|---|---|---|---|---|---|---|---|---|---|---|
| $g$ | BenQ (GA) | BenQ | NF4 | BenQ (GA) | BenQ | NF4 | BenQ (GA) | BenQ | NF4 | BenQ (GA) | BenQ | NF4 |
| 128 | 8616.16 | 13374.94 | 13525.77 | 1841.92 | 1981.44 | 1980.56 | 968.97 | 1029.68 | 1030.04 | 805.68 | 854.38 | 854.57 |
| 64 | 8746.31 | 13526.71 | 13436.66 | 1824.13 | 1981.15 | 1981.80 | 952.45 | 1030.06 | 1030.06 | 791.65 | 855.04 | 854.73 |

**Memory Reduction.** Tables 15 and 16 report the original (FP32) and 4-bit quantized memory footprints, along with the corresponding reduction ratios, for BenQ and BenQ (GA), respectively, across several model families.

BenQ consistently achieves higher compression than BenQ (GA): excluding embedding layers, the reduction ratios reach 7.55-7.76× for BenQ against 6.94-7.11× for BenQ (GA). This gap stems directly from the per-group overhead introduced by the group-adapted Benford logic, which stores additional scaling metadata for each quantization group.

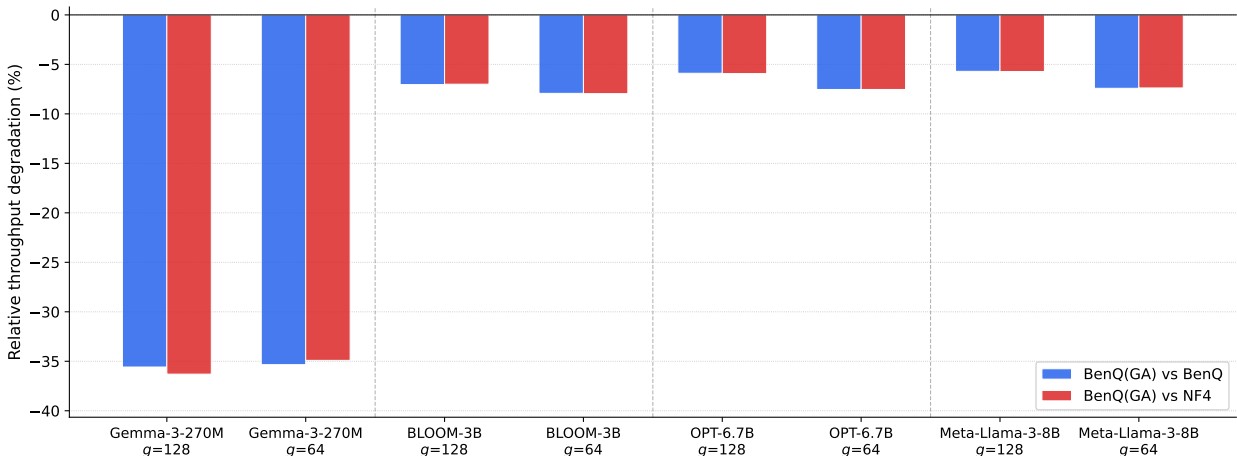

Figure 6: Relative throughput degradation of BENQ (GA) with respect to BENQ and NF4, across group sizes $g \in \{64, 128\}$ on perplexity task.

The overall reduction ratio varies considerably across model sizes, and is most affected by the relative weight of embedding layers. In small models such as Gemma-3-270M, embeddings account for over 90% of the quantized model size, yielding an overall ratio of only 1.48× despite strong compression of the remaining layers. Larger models, in which attention and feed-forward layers dominate, achieve ratios of up to 6.36× (BENQ on OPT-6.7B). Crucially, however, the reduction ratios computed excluding embeddings are remarkably stable across all evaluated models and both methods — ranging from 7.55× to 7.76× for BENQ — indicating that the compression of linear layers is consistent and largely independent of model scale.

For scenarios requiring more aggressive compression, quantizing embedding layers is a viable option: the component ablation study presented in Table 13 showed that including embeddings in the quantization process incurs only a marginal quality penalty, making it a favorable trade-off when memory constraints are rigorous.

Table 15: Memory reduction for BenQ quantization (4-bit, group size 128). Original size was calculated in FP32 precision.

|  | Gemma3-270M | Gemma3-1.0B | OPT-1.3B | OPT-2.7B | OPT-6.7B | Llama3-8.0B |
|---|---|---|---|---|---|---|
| Original size (MB) | 1,023 | 3,814 | 5,412 | 10,606 | 25,400 | 30,633 |
| Quantized size (MB) | 690 (emb: 640) | 1,496 (emb: 1,152) | 1,398 (emb: 786) | 2,243 (emb: 982) | 3,992 (emb: 786) | 7,441 (emb: 4,008) |
| Reduction ratio | 1.48× | 2.55× | 3.87× | 4.73× | 6.36× | 4.12× |
| Reduction ratio (w/o emb.) | 7.73× | 7.75× | 7.55× | 7.63× | 7.68× | 7.76× |

Table 16: Memory reduction for BenQ (GA) quantization (4-bit, group size 128). Original size was calculated in FP32 precision.

|  | Gemma3-270M | Gemma3-1B | OPT-1.3B | OPT-2.7B | OPT-6.7B | Llama3-8.0B |
|---|---|---|---|---|---|---|
| Original size (MB) | 1,023 | 3,814 | 5,412 | 10,606 | 25,400 | 30,633 |
| Quantized size (MB) | 694 (emb: 640) | 1,527 (emb: 1,152) | 1,452 (emb: 786) | 2,356 (emb: 982) | 4,280 (emb: 786) | 7,753 (emb: 4,008) |
| Reduction ratio | 1.47× | 2.50× | 3.73× | 4.50× | 5.93× | 3.95× |
| Reduction ratio (w/o emb.) | 7.09× | 7.10× | 6.94× | 7.00× | 7.04× | 7.11× |

**Quantizing Activations.** We extend our analysis to dynamic (or online) activation quantization, where quantization parameters are computed on the fly from activation statistics during each forward pass, rather than fixed through an offline calibration stage (Xiao et al., 2023), as discussed in Section 2.1, the spatial structure of neural networks promotes Benford-like behavior — making the log-uniform grid a favorable

candidate for representing activations. Once again, we adopt 4-bit quantization with a group size of 128. Tables 17, 18, and 19 report the results for the Qwen and Qwen3, BLOOM, and Gemma3 families, respectively. Furthermore, we evaluate BENQ under the activation smoothing mechanism of Xiao et al. (2023), with results reported on Tables 20 and 21.

Table 17: **4-bit activation quantization** with group size 128 on Qwen/Qwen3. Best quantized values are shown in bold.

| | Qwen-7B model | | | | | Qwen-14B model | | | | | Qwen3-4B model | | | | |
|---|---|---|---|---|---|---|---|---|---|---|---|---|---|---|---|
| | FP16 | BENQ (GA) | BENQ | NF4 | RTN | FP16 | BENQ (GA) | BENQ | NF4 | RTN | FP16 | BENQ (GA) | BENQ | NF4 | RTN |
| Lambada↑ | 0.639 | **0.601** | 0.555 | 0.568 | 0.501 | 0.354 | 0.418 | **0.422** | 0.325 | 0.317 | 0.635 | **0.557** | 0.522 | 0.539 | 0.463 |
| Perplexity↓ | 8.844 | **9.657** | 9.822 | 9.883 | 11.183 | 7.168 | **7.575** | 7.608 | 7.779 | 8.459 | 10.504 | **12.039** | 12.500 | 12.241 | 14.118 |

Table 18: **4-bit activation quantization** with group size 128 on BLOOM family. Best quantized values are shown in bold. † denotes divergence.

| | BLOOM-560M model | | | | | BLOOM-1.1B model | | | | | BLOOM-3.0B model | | | | |
|---|---|---|---|---|---|---|---|---|---|---|---|---|---|---|---|
| | FP16 | BENQ (GA) | BENQ | NF4 | RTN | FP16 | BENQ (GA) | BENQ | NF4 | RTN | FP16 | BENQ (GA) | BENQ | NF4 | RTN |
| HellaSwag↑ | 0.316 | **0.306** | 0.301 | 0.261 | 0.256 | 0.344 | 0.335 | 0.334 | **0.336** | 0.321 | 0.414 | **0.396** | 0.390 | 0.391 | 0.378 |
| Lambada↑ | 0.353 | **0.307** | 0.248 | 0.000 | 0.000 | 0.426 | **0.384** | 0.372 | 0.290 | 0.196 | 0.517 | **0.480** | 0.474 | 0.411 | 0.320 |
| MMLU↑ | 0.247 | **0.249** | 0.247 | 0.230 | 0.248 | 0.267 | 0.250 | **0.263** | 0.261 | 0.251 | 0.263 | **0.264** | 0.264 | 0.246 | 0.256 |
| Perplexity↓ | 23.326 | **32.910** | 38.985 | † | † | 18.465 | **21.627** | 22.175 | 26.472 | 41.620 | 13.991 | **15.642** | 16.270 | 18.824 | 26.317 |

Table 19: **4-bit activation quantization** with group size 128 on Gemma3 family. Best quantized values are shown in bold.

| | Gemma3-270M model | | | | | Gemma3-1.0B model | | | | |
|---|---|---|---|---|---|---|---|---|---|---|
| | FP16 | BENQ (GA) | BENQ | NF4 | RTN | FP16 | BENQ (GA) | BENQ | NF4 | RTN |
| HellaSwag↑ | 0.343 | **0.318** | 0.301 | 0.307 | 0.275 | 0.434 | **0.398** | 0.372 | 0.380 | 0.329 |
| Lambada↑ | 0.433 | **0.306** | 0.218 | 0.245 | 0.051 | 0.435 | **0.355** | 0.239 | 0.279 | 0.119 |
| MMLU↑ | 0.268 | **0.264** | 0.248 | 0.257 | 0.246 | 0.396 | **0.335** | 0.291 | 0.300 | 0.249 |
| Perplexity↓ | 24.453 | **44.364** | 74.880 | 57.352 | 267.430 | 28.757 | **42.379** | 57.837 | 49.968 | 118.125 |

Table 20: **4-bit activation quantization** with group size 128 and activation smoothing ($\alpha$=0.5) on BLOOM and Gemma3 families. Best values are shown in bold.

| | BLOOM-560M | | BLOOM-1.1B | | BLOOM-3B | | Gemma3-270M | | Gemma3-1.0B | |
|---|---|---|---|---|---|---|---|---|---|---|
| | BENQ (GA) | BENQ | BENQ (GA) | BENQ | BENQ (GA) | BENQ | BENQ (GA) | BENQ | BENQ (GA) | BENQ |
| HellaSwag↑ | **0.307** | 0.307 | **0.333** | 0.330 | **0.395** | 0.390 | **0.331** | 0.328 | **0.421** | 0.414 |
| Lambada↑ | **0.260** | 0.236 | **0.375** | 0.369 | 0.463 | **0.471** | **0.369** | 0.353 | **0.410** | 0.399 |
| MMLU↑ | **0.246** | 0.246 | **0.255** | 0.253 | **0.268** | 0.260 | 0.264 | **0.266** | **0.361** | 0.346 |
| Perplexity↓ | **32.526** | 36.658 | **22.022** | 22.430 | **15.694** | 16.050 | **30.308** | 32.697 | **32.988** | 35.433 |

Table 21: **4-bit activation quantization** with group size 128 and activation smoothing ($\alpha$=0.5) on Qwen3-4B. Best values are shown in bold.

| | BENQ (GA) | BENQ |
|---|---|---|
| Lambada↑ | **0.609** | 0.608 |
| Perplexity↓ | **11.142** | 11.205 |

**Discussion (activation quantization).** Across the evaluated settings, uniform RTN exhibits the largest degradation and can collapse on smaller models, confirming that naive uniform activation quantization is poorly suited to heavy-tailed activation distributions.

Non-uniform grids (NF4 and BenQ) can improve robustness over RTN, but results are mixed across model families. In BLOOM, BenQ avoids the divergence observed for NF4/RTN in some settings and yields substantially better downstream scores. In Gemma3-270M, however, all static-grid approaches remain challenging: BenQ reduces RTN collapse but still incurs large increases in perplexity, indicating that a fixed log-spaced grid is not universally sufficient for activations.

When equipped with smoothing (Xiao et al., 2023) ($\alpha$=0.5), both BenQ variants exhibit consistent and substantial improvements on Gemma3 and Qwen3 families across all metrics, as reported in Tables 20 and 21. For instance, on Gemma3-270M, BenQ (GA) reduces perplexity from 44.36 to 30.31 and raises Lambada accuracy from 0.306 to 0.369. BenQ benefits even more markedly, dropping perplexity from 74.88 to 32.70 and improving Lambada from 0.218 to 0.353.

These results suggest that, for these model families, the difficulty of 4-bit dynamic activation quantization stems largely from outlier activations rather than from an inadequacy of the log-uniform grid itself: once the activation scale is migrated to the weights via smoothing, the quantization grid operates on a better-conditioned distribution and recovers much of the lost quality. Also about these models, the performance gap between BenQ and BenQ (GA) narrows considerably under smoothing, indicating that group-wise granularity becomes less critical when the input distribution is pre-conditioned.

In contrast, the BLOOM family does not benefit from smoothing under $\alpha$=0.5: results remain largely unchanged or degrade slightly across models. On BLOOM-1.1B, for instance, BenQ (GA) sees Lambada accuracy drop from 0.384 to 0.375 and perplexity increase from 21.63 to 22.02. A similar pattern holds for BenQ, with Lambada falling from 0.372 to 0.369 and perplexity rising from 22.18 to 22.43.

This behavior contrasts sharply with the Gemma3 and Qwen3 families and suggests that $\alpha$=0.5 is not a universally appropriate choice: the BLOOM architecture — characterized by ALiBi positional encoding and pre-LayerNorm — may produce activation distributions that are already sufficiently well-conditioned, such that migrating scale to the weights via smoothing offers no benefit and may introduce unnecessary distortion. Adapting $\alpha$ per model family, or performing a light per-layer search, remains a promising direction for making activation smoothing more robust across architectures.

## 5 Limitations

This study has limitations that are important for interpreting the results.

First, BenQ uses a simple static log-spaced grid motivated by Benford-like scale-broad statistics; we do not claim that this grid is optimal for any specific analytic prior, nor do we provide a proof of optimality (e.g., via quantile-based derivations or rate–distortion arguments). Accordingly, BenQ should be viewed as a lightweight baseline or component rather than a universal replacement for optimized PTQ methods.

Second, although logarithmic quantization levels are known to enable hardware-friendly implementations – particularly through the replacement of multiplications with bit-shift operations (Lee et al., 2017; Przewlocka-Rus et al., 2022) – we do not evaluate BenQ in this regard. Our experiments measure quantization quality under standard inference pipelines, without optimized kernels or hardware-level profiling of energy consumption or throughput. Whether the log-spaced grid of BenQ translates into practical inference acceleration or energy efficiency gains remains an open question and an important direction for future work.

Third, comparisons to optimization- and activation-aware methods (e.g. GPTQ/AWQ/SINQ) are included to situate BenQ relative to stronger calibration/optimization-based baselines. We do not claim that a static, data-free codebook should systematically outperform such methods, and we do not exhaustively tune their hyperparameters beyond standard group size.

Finally, our activation quantization experiments are conducted with a fixed smoothing coefficient ($\alpha$=0.5), without per-layer or per-architecture tuning. We therefore treat the activation results as indicative rather

than exhaustive, and expect that a more systematic exploration of complementary mechanisms (*e.g.* smoothing, clipping, calibration) would better characterize the practical ceiling of log-uniform activation quantization.

## 6 Conclusion and Future Work

We conducted this study to analyze Benford's Law as a distributional prior for post-training quantization of LLMs. Initially, we presented the theoretical formulation that justifies the expectation of Benford adherence in model weights (and activations). More specifically, Benford adherence was expected for `nn.Linear` layers (MLP/Attention/FeedForward), whereas normalization layers were not expected to exhibit such behavior. Subsequently, based on the MAD metric (Equation 15) and the $\epsilon$-Benford adherence test (Campanelli, 2022), we carried out an empirical evaluation of this hypothesis, which was confirmed — as illustrated in Figure 3 and Table 1. Four out of six evaluated models were found to be 20%-Benford adherent, with the two exceptions driven by the influence of normalization layers on the model-wise test. We observed that the level of Benford adherence may vary across models, which can be explained by the quality of the training process/data and intrinsic architectural differences.

Subsequently, we proposed BenQ, a PTQ method inspired by Benford's Law that combines selective quantization (i.e., quantizing only `nn.Linear` layers while excluding normalization and embedding layers) with a log-uniform grid. The method was evaluated against RTN, NF4, GPTQ, AWQ and SINQ. It was observed that non-uniform grids frequently outperformed the uniform baseline, suggesting that non-uniform grids can allocate quantization precision more effectively than uniform RTN in these settings. However, when comparing BenQ with NF4, it was not possible to establish a relationship of absolute superiority, as the optimal grid choice depends on the specific architecture and downstream task under consideration. Finally, when situated against state-of-the-art PTQ methods (GPTQ/AWQ/SINQ), BenQ is generally weaker in perplexity—as expected for a static, data-free codebook. This comparison is included primarily to situate it relative to stronger baselines rather than as a claim of superiority, with BenQ serving as a simple, data-free alternative or hybrid component.

We also conducted parameter sweep over bit-widths and group sizes, and a ablation study over BenQ's core design choices. The component ablation revealed an asymmetry in the selective quantization policy: excluding normalization layers yields a far greater quality improvement than excluding embedding layers, confirming that preserving `LayerNorm` precision is the dominant factor. Notably, the limited quality penalty of quantizing embeddings carries practical implications for memory-constrained deployments: since embedding layers account for a large fraction of total model size in smaller LLMs, being able to quantize them with minimal accuracy loss would make BenQ more effective precisely where memory savings matter most.

Finally, we report dynamic activation quantization results for BenQ. While BenQ consistently improves over RTN and can be more robust than NF4 in specific families (e.g., BLOOM), performance is mixed across models. We also evaluated BenQ combined with activation smoothing (Xiao et al., 2023), which substantially improves quality on Gemma3 and Qwen3 families, suggesting that outlier activations were the main bottleneck rather than the log-uniform grid itself. Overall, reliable low-bit activation PTQ likely requires such complementary mechanisms beyond a static grid.

Future work includes analyzing BenQ on models trained with Benford regularization (Ott et al., 2025), since the method is naturally aligned with such settings and this form of regularization may further enhance the effectiveness of log-uniform quantization. Another promising direction is the hybridization of BenQ with state-of-the-art methods (e.g., AWQ, GPTQ, SINQ) as well as recent multi-stage compression frameworks such as TurboQuant (Zandieh et al., 2025). In this context, BenQ could act as a lightweight first-stage quantizer for tensors exhibiting Benford-like, scale-broad statistics, while a second residual- or inner-product-preserving stage could compensate for the limitations of a static log-spaced grid in more sensitive scenarios, particularly for activations and cache-like representations. Finally, extending the analysis of BenQ to architectures of different natures—such as Vision Transformers and TSFMs—remains an important direction for future work.

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
