# OpenReview forum: "Benford’s Law as a Distributional Prior for Post-Training Quantization of Large Language Models"
_TMLR — Decision pending for TMLR_

### Review · Reviewer_Vbx9 · 2026-05-03

**Summary Of Contributions:**

This paper studies the post-training quantization (PTQ) method from the perspective of Benford's Law. The paper provided both conceptual explanation and experimental results to support the argument that parameters certain layers in LLMs has the property that the leading digits follow the Benford's Law. Based on this observation, the paper proposed the BenQ and GA-BenQ method that partitions the normalized space into logarithmically even intervals, and round the digits to the nearest anchor. The paper also identified that certain layers does not always follow the Benford's Law, so quantization to those layers are skipped. The paper provided experimental result to compare the proposed methods with both simple quantization methods (NF4 and RTN), and more complicated methos that involves additional optimization and calibration (GPTQ and AWQ), and demonstrates strong performance.

**Strengths**
1. Paper is well-written, with a clear motivation of the proposed method, and good explanation of the method it self.
2. Experiments conducted on a number of interesting LLMs.

**Weaknesses**
1. The connection between the empirically observed Benford Law and the proposed interval partition is weak. In particular, the Benford's Law (and what's verified in the experiments) are describing the property of the leading digit, while the quantization algorithm use a log-even partition of the whole interval (\epsilon, 1].
2. The paper lacks a thorough discussion about how to choose the groups in the quantization algorithm.
3. Saving the group minimum and maximum might incur additional cost, which is not discussed in the paper.

**Audience:**

Yes

**Audience Explanation:**

Post-training quantization is an important topic that can make the models run on less powerful devices.

**Broader Impact Concerns:**

None.

**Claims And Evidence:**

Yes

**Claims Explanation:**

Yes. The paper claims am empirical verification of Bernford's Law, the BenQ codebook, a selective quantization strategy, and an emiprical study of the proposed methods. All claimed contribution are presented in the paper.

**Requested Changes:**

1. A better explanation (maybe with an extended empirical diagnostic) of the empirically observed Benford Law that goes beyond the leading digit.

2. A discussion about how to choose the groups in the quantization algorithms and how storing the group min and max influence the quantiztion efficiency.

3. A discussion or clarification of the computation cost of inference of models with this quantization.

---

> ### Author Response · Authors · 2026-05-15
>
> First of all, thank you for your feedback and suggestions, which helped improve the work. A revised version of the paper and code is already available on OpenReview. Regarding the specific changes requested, we have organized the following modifications to the paper:
>
> 1. A better explanation (maybe with an extended empirical diagnostic) of the empirically observed Benford Law that goes beyond the leading digit.
>
> We extended our empirical diagnostic to the second digit, as seen in Figure 3 of the revised paper. The second-digit analysis yielded results consistent with the first-digit findings: LayerNorm layers consistently exhibited the highest MAD values across all evaluated models, while transformational layers remained close to the generalized Benford distribution. The dichotomy between layer types was preserved, despite the absolute MAD values being smaller due to the larger number of categories (90 pairs vs. 9 digits). Notably, the LayerNorm outliers observed in the Gemma family under the first-digit analysis were substantially reduced in the second-digit analysis, suggesting that the deviations in that family are more concentrated in the first significant digit than in the joint two-digit distribution. Overall, this consistency across both digit orders strengthens the evidence that the Benford-like behavior of transformational layers reflects a broader logarithmic regularity in the weight magnitudes, rather than an artifact of the first-digit distribution alone.
>
> Beyond the visual and descriptive analysis, we also strengthened the statistical rigor of the first-digit analysis by adopting Campanelli's epsilon-Benford adherence test, which is asymptotically independent of sample size. The results are presented in Table 1 of the revised paper.
>
> 2. A discussion about how to choose the groups in the quantization algorithms and how storing the group min and max influence the quantiztion efficiency.
>
> We addressed this point in two ways in the revised paper.
>
> First, we added a group-size ablation study (Tables 10 and 11 of the revised paper), which analyzed the effect of group sizes 32, 64, 128, and 256 on quantization quality. The results showed that smaller groups tend to improve fidelity by better aligning the quantization grid with the local weight distribution, but at the cost of increased metadata overhead — a trade-off that is particularly pronounced for BenQ (GA), which stores per-group min and max values in addition to the standard scale factor. Based on these results, we recommend group size 32 when accuracy is prioritized, group size 64 as a strong practical trade-off between performance and compression, and we retained group size 128 as the standard setting for fair comparison with common PTQ configurations.
>
> Second, we added a memory reduction analysis (Tables 13 and 14 of the revised paper), which quantifies how the per-group metadata overhead of BenQ (GA) impacts the effective compression ratio relative to BenQ: excluding embedding layers, BenQ achieves reduction ratios of 7.55–7.76x, while BenQ (GA) reaches 6.94–7.11x, with the gap stemming directly from the additional min/max storage per group. This analysis makes the efficiency cost of group-adaptive quantization explicit and allows practitioners to make informed decisions based on their memory and accuracy constraints.
>
> 3. A discussion or clarification of the computation cost of inference of models with this quantization.
>
> Our current implementation does not include optimized CUDA kernels or hardware-level profiling, and therefore we do not report inference throughput or latency measurements. We acknowledge this as a limitation, and we explicitly discuss it in the limitations section of the revised paper, identifying it as an important direction for future work. We note, however, that prior work in the literature has demonstrated that logarithmic quantization levels can be aligned with hardware acceleration — particularly through the replacement of multiplications with bit-shift operations — suggesting that BenQ's log-spaced grid may carry practical inference efficiency advantages. Nevertheless, whether these translate into measurable gains in our specific setting remains an open question that we do not claim to have answered.
>
> As a partial characterization of computational efficiency, we report the memory footprint of BenQ and BenQ (GA) in Tables 13 and 14 of the revised paper, which show substantial reductions in memory usage relative to full-precision models.

---

### Review · Reviewer_Aumc · 2026-05-04

**Summary Of Contributions:**

This paper proposes BenQ, a data-free post-training quantization (PTQ) method for LLMs that combines (1) a log-spaced quantization grid motivated by Benford's Law and (2) a selective quantization strategy that quantizes only linear layers while keeping others (LayerNorm and embedding) in higher precision. The paper also introduces BenQ-GA, a group-adaptive variant that generates a per-group log-spaced grid using each group's actual min/max values instead of a normalized [-1, 1] grid with per-group scaling. The core empirical observation is a functional dichotomy in transformer weights: across multiple model families (Llama3, Gemma3, Qwen, BLOOM, OPT), the paper observes that the linear weights closely follow Benford's leading-digit distribution. The paper provides a log-domain rationale for this observation based on multiplicative training dynamics (SGD with weight decay and adaptive preconditioning) and the spatial structure of repeated matrix products. The paper evaluates 4-bit weight-only PTQ with group size 128 on perplexity (WikiText-2), HellaSwag, LAMBADA, and MMLU, comparing BenQ against RTN, NF4, GPTQ, and AWQ. It additionally reports activation quantization results as an exploratory stress test.

### Key strengths

**S1**: The diagnostic observation about the Linear-vs-LayerNorm Benford dichotomy is well-supported visually (Figure 1) and quantitatively via MAD (Figure 2).

**S2**: The selective quantization strategy is principled and motivated directly by the observations.

### Key weaknesses

**W1**: The empirical advantage over the baseline is small and sometimes negative. Across Tables 1-7, BenQ is outperformed by baselines such as NF4 and AWQ in many model-metric configurations. Moreover, some configurations, such as BenQ on the Qwen3-4.0B model on perplexity, exhibit significant accuracy degradation.

**W2**: Evaluation is limited to a single bit-width (4-bit) and a single group size (128); the theoretical motivation predicts that log spacing should help most at lower bit-widths (2-3 bits), but this is not tested.

**W3**: No comparison against block-scaling FP formats such as MXFP4 or NVFP4, which represent the practical 4-bit deployment frontier in modern hardware like Blackwell.

**Audience:**

Yes

**Audience Explanation:**

The contribution on Benford's law is novel in the LLM quantization context and could be of interest to subcommunities within TMLR's audience, including researchers working on LLM efficiency and PTQ, as well as those studying neural network weight distributions and training dynamics.

**Broader Impact Concerns:**

There are no broader impact concerns.

**Claims And Evidence:**

No

**Claims Explanation:**

The claim about Transformer weights with respect to Benford's law is well supported by Figures 1 and 2 across six models from four families and is consistent with classical Benford theory. However, the claim about the effectiveness of BenQ as a PTQ method is not adequately supported, for three main reasons:

1. Across Tables 1-3, NF4 and BenQ/BenQ-GA trade wins roughly evenly, with NF4 sweeping all four metrics on Llama3-8B. On Qwen3-4B (Table 4), plain BenQ achieves a perplexity of 17.22, which is significantly worse than BenQ-GA and other methods. The activation results (Tables 5-7) are similarly inconsistent: BenQ helps on BLOOM but degrades severely on Gemma3-270M.
2. The paper evaluates only 4-bit at group size 128. Other bit-widths (e.g., 2-bit, 3-bit, 8-bit) are not tested, so it is unclear how BenQ behaves outside this single configuration. The comparison also omits block-scaling FP formats (MXFP4, NVFP4) that represent the practical 4-bit deployment frontier on modern hardware (Blackwell).
3. The MAD analysis lacks formal goodness-of-fit testing (chi-squared, Kolmogorov-Smirnov) or null-distribution baselines; the "Benford-like" label is asserted by visual inspection alone.

**Requested Changes:**

### Critical

1. Evaluate BenQ at other bit-widths (e.g., 2-bit, 3-bit, 8-bit) in addition to 4-bit. The current single-bit-width evaluation does not show how BenQ behaves outside this one configuration.
2. Add comparisons against block-scaling FP formats (MXFP4, NVFP4). These are the practical 4-bit formats used on modern hardware such as Blackwell GPUs, and their empirical positioning is incomplete without them.
3. Add formal Benford goodness-of-fit testing (e.g., chi-squared or Kolmogorov-Smirnov) alongside the MAD analysis. The "Benford-like" label is currently asserted based on visual inspection of MAD values, without any statistical test or null baseline.
4. Explain the Qwen3-4B perplexity collapse. Plain BenQ degrades to 17.22 on Qwen3-4B (Table 4) vs. BenQ-GA at 11.77 and GPTQ/AWQ at ~11. This robustness anomaly is not analyzed in the text.
5. Reframe or strengthen the comparison against NF4. Across Tables 1-3, NF4 and BenQ trade wins roughly evenly, with NF4 sweeping all metrics on Llama3-8B. Either demonstrate a setting where BenQ clearly outperforms NF4, establish a criterion for when log-spaced grids should be preferred, or reframe the contribution as a diagnostic plus lightweight alternative rather than a competitive method.

### To strengthen the work

1. Add a group-size ablation (e.g., 32, 64, 128, 256). The interaction between group size and codebook shape is non-trivial, and tighter group ranges may erode BenQ's advantage.
2. Expand the activation quantization experiments with outlier-handling mechanisms (smoothing, clipping, rotation) to evaluate BenQ's applicability in weight-activation quantizations.
3. Add an ablation isolating the two components of BenQ (log-spaced grid vs. selective quantization). This would clarify which aspect drives the observed gains.
4. Add comparisons to additional recent SOTA PTQ methods (e.g., SqueezeLLM, OmniQuant, Atom, QuIP#, AQLM, SpinQuant, HQQ) to better situate the practical relevance of BenQ.
5. Provide measured hardware efficiency numbers (throughput or energy) to support the hardware-friendliness claim.

---

> ### Author Response · Authors · 2026-05-15
> **Response (part 1/3)**
>
> First of all, thank you for your feedback and suggestions, which helped improve the work. A revised version of the paper and code is already available on OpenReview. Due to the character limit per response on OpenReview, we have split our response into 3 comments.
>
> Regarding the specific changes requested, we have organized the following modifications to the paper:
>
> Critical:
> 1. Evaluate BenQ at other bit-widths (e.g., 2-bit, 3-bit, 8-bit) in addition to 4-bit. The current single-bit-width evaluation does not show how BenQ behaves outside this one configuration.
>
> We evaluated BenQ across four bit-widths, 2-, 3-, 4-, and 8-bit, and further compared it against the RTN baseline. The results are presented in Tables 8 and 9 of the revised version of the paper. We pointed out that BenQ, especially in its group-adapted version, consistently allocates precision more effectively than RTN in low-bit regimes. However, in the 8-bit regime, RTN achieved better performance, which is consistent with the expectation that uniform quantization suffices when the number of levels (2^8 = 256) is large enough to cover the weight distribution without significant error. Finally, we also showed that the 2-bit scenario still poses a challenge for both methods across all evaluated models.
>
> 2. Add comparisons against block-scaling FP formats (MXFP4, NVFP4). These are the practical 4-bit formats used on modern hardware such as Blackwell GPUs, and their empirical positioning is incomplete without them.
>
> We compared BenQ against NVFP4 and MXFP4 (see Tables 6 and 7 of the revised paper), with BenQ achieving better results than both methods. We attributed this to two key differences: first, BenQ (GA) employs adaptive log-uniform levels per group, fitted to the actual value range of each weight block, whereas NVFP4 and MXFP4 rely on the fixed E2M1 format; and second, BenQ stores scale factors in FP16, while NVFP4 and MXFP4 constrain their scales to 8 bits — E4M3 and E8M0, respectively. We also pointed out that, however, this comparison is not entirely fair: NVFP4 and MXFP4 are formats designed with hardware efficiency as a primary concern — with native support in NVIDIA's Blackwell Tensor Cores — which imposes representational constraints in exchange for gains in throughput and energy efficiency, and BenQ, in turn, carries no such hardware commitment. Thus, we stated that these results reflect quantization quality under idealized conditions and should not be extrapolated to real-world inference scenarios.
>
> 3. Add formal Benford goodness-of-fit testing (e.g., chi-squared or Kolmogorov-Smirnov) alongside the MAD analysis. The "Benford-like" label is currently asserted based on visual inspection of MAD values, without any statistical test or null baseline.
>
> Benford goodness-of-fit testing is not a straightforward task in LLM weights context. In the revised version of the paper, we showed, in Figure 2, that even small deviations from BL incurs in strong rejection of null hypothesis in chi squared test. Cerqueti & Lupi (2023) explain that this phenomenon is common for many statistical tests when the sample size is very large because, as the sample size grows, statistical tests gain increasing power to detect even negligible deviations from the null hypothesis — deviations that are practically meaningless from a substantive standpoint.
>
> To address this problem, we adopted Campanelli’s test (also known as epsilon-benford adherence test), which is asymptotically independent of sample size. It relaxes the notion of benford adherence through a parameter epsilon, which allows us to assess, with statistical rigor, whether the relative deviation of each digit $d$ with respect to its Benford probability is bounded by it. The results of this test are displayed in Table 1 of the revised paper, and they revealed that 4 out of 6 analysed models were 20%-Benford adherent (model-wise). These findings were further supported by the layer-wise MAD analysis, already present in the previous version of the paper, which provided complementary evidence that the observed rejections are largely driven by LayerNorm layers, reinforcing the motivation for the selective quantization policy.
>
> However, when developing the code for Campanelli’s test, we noticed a bug in an auxiliary script we had previously developed for analysing the massive layer-wise data about the MAD metric. It was wrongfully considering “Embedding Normalization Layers” as “Embedding Layers”, which led us to make a wrong claim that Embedding layers do not adhere to Benford’s Law when, in reality, normalization layers were the only ones that strongly deviated. In face of that, we removed this incorrect claim in the paper and we also regenerated the figure which depicts the MAD analysis by layers (Figure 3 of the revised paper version), confirming that the embedding layers were not in the highly deviating group of layers (the ones in the left of each subfigure).
>
> [Continues on Response (part 2/3)]

---

> > ### Author Response · Authors · 2026-05-15
> > **Response (part 2/3)**
> >
> > Critical:
> >
> > 3. Continuation...
> >
> > Despite this correction, we maintained the decision to exclude embedding layers from quantization in the selective quantization policy, motivated by the goal of preserving vocabulary representational capacity, given that embedding layers serve as the interface between discrete tokens and the continuous representation space. This is also consistent with the default behavior of widely used PTQ frameworks — for instance, HqqConfig, HiggsConfig, and FourOverSix in the Transformers library exclude lm_head by default, and BitsAndBytesConfig explicitly recommends keeping it in its original precision. In most LLMs, lm_head is weight-tied to the input embedding layer, meaning both refer to the same underlying parameter matrix. Also, the effects of quantizing embedding layers are further explored in our component ablation study, discussed in Item 3 of the 'To Strengthen the Work' section of this rebuttal.
> >
> > 4. Explain the Qwen3-4B perplexity collapse. Plain BenQ degrades to 17.22 on Qwen3-4B (Table 4) vs. BenQ-GA at 11.77 and GPTQ/AWQ at ~11. This robustness anomaly is not analyzed in the text.
> >
> > This perplexity collapse was, in reality, a LaTeX table formatting error. The actual perplexity for this configuration is 11.772, which does not constitute an anomaly. After detecting this error, we carefully reviewed all tables and identified two additional formatting mistakes: in Tables 3 and 5 of the original version of the paper (OPT's weights table and Qwen's activation table, respectively), the column headers of BenQ and BenQ (GA) were inadvertently swapped. It is important to emphasize that neither the results nor the analyses were affected by these errors, as all experiments were originally inspected using a separate data management tool — the incorrect labels were introduced only during the LaTeX formatting stage.
> >
> > 5. Reframe or strengthen the comparison against NF4. Across Tables 1-3, NF4 and BenQ trade wins roughly evenly, with NF4 sweeping all metrics on Llama3-8B. Either demonstrate a setting where BenQ clearly outperforms NF4, establish a criterion for when log-spaced grids should be preferred, or reframe the contribution as a diagnostic plus lightweight alternative rather than a competitive method.
> >
> > We agree that no consistent superiority is observed in either direction: both methods trade wins across models and metrics, with differences typically within 1-2 percentage points. These results suggest that, at 4-bit group-wise PTQ, the choice between log-spaced and normal-quantile static grids is architecture- and task-dependent, with neither dominating universally. We therefore reframed BenQ's contribution in the revised paper (see Section 4.2, “Discussion” topic): rather than positioning it as a competitive method against NF4, we present it as a diagnostic-motivated, data-free lightweight alternative, whose design is directly grounded in the Benford adherence analysis, providing an empirically-motivated justification for log-spaced grids in LLM quantization.
> >
> > To strengthen the work:
> >
> > 1. Add a group-size ablation (e.g., 32, 64, 128, 256). The interaction between group size and codebook shape is non-trivial, and tighter group ranges may erode BenQ's advantage.
> >
> > We conducted a group-size ablation of BenQ and BenQ (GA), as presented in Tables 10 and 11 of the revised paper. Overall, smaller group sizes tended to improve quantization fidelity, with the effect being particularly pronounced for Gemma models. These results indicated that fine-grained grouping improved the local alignment between the quantization grid and the empirical weight distribution, as larger groups aggregated more heterogeneous values under a single scale or range, increasing reconstruction error. This effect was especially visible for BenQ (GA), whose per-group grid directly benefited from accurately capturing local weight ranges. However, smaller groups also increased metadata overhead, introducing a memory-quality trade-off. We therefore retained group size 128 as the standard setting for comparison with common group-wise PTQ configurations, while the ablation demonstrated that BenQ is not tied to this single operating point
> >
> > 2. Expand the activation quantization experiments with outlier-handling mechanisms (smoothing, clipping, rotation) to evaluate BenQ's applicability in weight-activation quantizations.
> >
> > We expanded our activation quantization experiments with the smoothing mechanism (α=0.5), as presented in Tables 18 and 19 of the revised paper. For Gemma3 and Qwen3 families, both BenQ variants exhibited consistent and substantial improvements across all metrics. These results suggested that the difficulty of 4-bit dynamic activation quantization in these models stemmed largely from outlier activations rather than from an inadequacy of the log-uniform grid itself. In contrast, the BLOOM family did not benefit from smoothing under α=0.5, with results remaining largely unchanged or degrading slightly.

---

> > > ### Author Response · Authors · 2026-05-15
> > > **Response (part 3/3)**
> > >
> > > To strengthen the work:
> > >
> > > 2. Continuation...
> > >
> > > This contrasting behavior suggested that α=0.5 is not a universally appropriate choice, and that the BLOOM architecture may already produce sufficiently well-conditioned activation distributions, such that smoothing offers no benefit and may introduce unnecessary distortion. We noted that adapting α per model family, or performing a light per-layer search, remains a promising direction for making activation smoothing more robust across architectures.
> > >
> > > 3. Add an ablation isolating the two components of BenQ (log-spaced grid vs. selective quantization). This would clarify which aspect drives the observed gains.
> > >
> > > We conducted an ablation study on BenQ's core components — the logarithmic quantization levels and the selective layer quantization policy —, as presented in Table 12 and Figure 5 of the revised paper. The results revealed that excluding normalization layers from quantization yielded a more significant quality improvement than excluding embedding layers alone, suggesting that normalization layers are disproportionately harmful when quantized and that preserving their precision is the dominant factor in the selective quantization policy. Regarding the interaction between quantization levels and layer type, we observed an important asymmetry: while logarithmic levels outperformed uniform ones for nn.Linear layers, the opposite held for normalization layers — a finding consistent with the Benford adherence analysis, as normalization layer weights deviate strongly from Benford's Law, making the log-uniform grid a poor fit for their distribution. Finally, we also noticed that the advantage of logarithmic levels over uniform ones was not unconditional: it only materialized when non-Benford adherent layers were excluded from quantization. This indicated that the two components are interdependent — the selective quantization policy is not merely an auxiliary design choice, but a necessary condition for the log-uniform grid to express its representational advantage.
> > >
> > > 4. Add comparisons to additional recent SOTA PTQ methods (e.g., SqueezeLLM, OmniQuant, Atom, QuIP#, AQLM, SpinQuant, HQQ) to better situate the practical relevance of BenQ.
> > >
> > > We extended our comparisons of BenQ against SOTA PTQ methods with SINQ, as seen in Table 5 of the revised paper. SINQ is a recent calibration-free method that introduces a second-axis scale factor to the weight matrix, using a Sinkhorn-Knopp-style algorithm to iteratively normalize per-row and per-column weight variances — which, as the authors showed, approximates activation-aware quantization without requiring calibration data. Its authors claim it outperforms SOTA methods such as HQQ, GPTQ, and AWQ across several settings. Overall, SINQ achieved competitive results across both evaluated models, attaining the best metrics on Qwen3-4B model.
> > >
> > > 5. Provide measured hardware efficiency numbers (throughput or energy) to support the hardware-friendliness claim.
> > >
> > > In the revised version of the paper, we removed the hardware-friendliness claim, as we do not provide measured throughput or energy efficiency numbers to support it — our experiments evaluate quantization quality exclusively under standard inference pipelines, without optimized CUDA kernels or hardware-level profiling. We note, however, that the potential for hardware-friendly implementations of logarithmic quantization is well-supported in the literature, as log-spaced grids can replace multiplications with bit-shift operations, enabling more efficient hardware execution. Nevertheless, whether BenQ's log-spaced grid translates into practical inference acceleration or energy efficiency gains remains an open question, which we explicitly acknowledge in the limitations section of the revised paper as an important direction for future work.
> > >
> > > However, we added data about the memory reduction BenQ and BenQ (GA) promoted (Tables 13 and 14 of the revised paper). Although they are not enough to support the hardware-friendliness claim alone, they revealed that BenQ and BenQ (GA) are capable of promoting substantial reductions in memory usage. Also, the data revealed that the reductions can be even higher when we quantize embedding layers: especially in smaller LLMs, non-quantized embedding layers can represent more than 90% of the memory used by the quantized model.

---

### Review · Reviewer_u1Z6 · 2026-06-12

**Summary Of Contributions:**

The paper introduces BenQ, a Post-Training Quantization (PTQ) method motivated by Benford-like statistics of Linear layer parameters. BenQ is a codebook-based, data-free quantization approach establishing log-spaced quantization levels that are applied group-wise on weights (primarily) and/or activations.

The authors motivate the emergence of Benford-like statistics, and investigate the impact of 4-bit BenQ (and a Group-Adaptive variant, BenQ(GA)) on various model families (Gemma, Llama, OPT, BLOOM, Qwen) with parameters ranging from 270M to 14B. BenQ is benchmarked against competing data-free approaches (RTN, NF4, NVFP4/MXFP4) and compared to calibration-based PTQ techniques (GPTQ, AWQ, SINQ). Results show BenQ is competitive against selected data-free approaches.

The paper is easy to follow and understand, although sometimes redundant in its exposition. It presents convincing experiments demonstrating that Linear layers tend to adhere to Benford's law, while LayerNorm and Embeddings do not (with some interesting variations depending on model family). The motivations for the emergence of such behavior in Linear layers (Section 2.1) are also convincing. References are appropriate.

**Additional Comments:**

n/a

**Audience:**

No

**Audience Explanation:**

The adherence of Linear layer parameters to Benford's law is of interest but previously reported in the literature.
The proposed algorithm doesn't appear to meaningfully improve performance over competing techniques and, as such, is unlikely to find practical application.

**Broader Impact Concerns:**

Not addressed in the manuscript. No concerns on my end.

**Claims And Evidence:**

No

**Claims Explanation:**

1. Flawed comparisons against other techniques: the proposed method is a group-wise quantization approach with logarithmically-spaced levels which closely resemble conventional group-wise floating-point (FP) quantization. In a sense, the method corresponds to a non-base2 floating point method with zero mantissa (with implications for lower hardware compatibility and more computational overheads than FP). This connection should have been drawn more clearly. In this regard, in Section 4.2 a comparison is drawn with recent microscaling formats (NVFP4, MXFP4); as correctly remarked in the text, the direct comparison is unfair because these formats employ a quantized scaling factor, as well as a two-level scaling in the case of NVFP4. If the purpose of the comparison was to establish whether BenQ selection of quantization levels can improve (even without claims of optimality) on existing techniques that use non-uniform grids, it would have been more meaningful to present a comparison against a simplified FP4 E2M1 format with unquantized scales. Does BenQ grid selection shows any advantage against such logarithmic scale?

2. Unclear motivations for BenQ: when compared to other data-free methods with non-uniform grids, BenQ accuracy/perplexity are very comparable with NF4, and potentially to FP4 E2M1 or other non-uniform grids. In the scenarios when these methods show better performance than integer quantization, this is usually attributed to the good handling of outliers while preserving near-zero representation. These are well known and studied effects. If BenQ grid doesn't meaningfully improve representation, it is not clear what's the motivation of using yet another log grid, just with different spacing.

3. Incomplete characterization of BenQ variant: BenQ(GA) variant defines an individual set of quantization levels for each group. The additional quantization metadata impact compression (as shown in Fig. 4 and Table 14). Does BenQ(GA) has compute overheads? What is the runtime of BenQ(GA) vs BenQ vs NF4 for a sizeable model?

4. Hyperparameter selection clarification: equations 10 and 11 define the BenQ codebook levels. These levels depend strongly on the choice of epsilon. How was epsilon selected? How does it impact performance?

5. Misleading claims about layer-selective quantization: it is repeatedly remarked that Linear layers are quantized with BenQ while LayerNorm and Embeddings are kept in high precision because the former adhere to Benford's Law, while the latter do not. It is not mentioned however that is also the case for the vast majority of quantization algorithms for model deployment (one sentence in Section 4.2 seems to imply otherwise: "In contrast, the models ... framework."). It is not clear then if the increased degradation upon LayerNorm and/or Embeddings quantization (Figure 5) is driven by a failure that is specific to BenQ and lack of adherence to Benford's Law. Similar behaviors are observed under integer quantization.

**Requested Changes:**

Please address point 1-5 above.

Additional minor comment: use of the term "ablation study" is often imprecise. When used for Bit-width and Group-size (Section 4.2), it refers to a parameter sweep, not to a study where a component is removed / disabled.

---

> ### Author Response · Authors · 2026-06-26
> **Response (part 1/3)**
>
> First of all, thank you for your feedback and suggestions, which helped improve the work. A revised version of the paper and code is already available on OpenReview.
>
> Regarding the specific changes requested, we have organized the following modifications to the paper:
>
> 1. Flawed comparisons against other techniques: the proposed method is a group-wise quantization approach with logarithmically-spaced levels which closely resemble conventional group-wise floating-point (FP) quantization. In a sense, the method corresponds to a non-base2 floating point method with zero mantissa (with implications for lower hardware compatibility and more computational overheads than FP). This connection should have been drawn more clearly. In this regard, in Section 4.2 a comparison is drawn with recent microscaling formats (NVFP4, MXFP4); as correctly remarked in the text, the direct comparison is unfair because these formats employ a quantized scaling factor, as well as a two-level scaling in the case of NVFP4. If the purpose of the comparison was to establish whether BenQ selection of quantization levels can improve (even without claims of optimality) on existing techniques that use non-uniform grids, it would have been more meaningful to present a comparison against a simplified FP4 E2M1 format with unquantized scales. Does BenQ grid selection shows any advantage against such logarithmic scale?
>
> Following your suggestion, we added Table 6 in the revised paper, which compares BenQ against a standard FP4 E2M1 baseline with unquantized FP32 scales across 3 different models.
> The results show that BenQ is competitive with this FP4 baseline and improves upon it in several model-metric configurations. The most notable gain appears on Gemma3-1B, where BenQ (GA) reduces perplexity from 41.95 to 35.18 in relation to FP4. On TinyLlama-1.1B, BenQ achieves lower perplexity (8.60 vs. 8.79) despite FP4 holding a small advantage on HellaSwag. On Llama3-8B, BenQ is competitive across HellaSwag and LAMBADA, and shows more pronounced advantages on MMLU and perplexity, where BenQ achieves 0.634 and 6.984 respectively, compared to 0.579 and 7.156 for FP4. These findings suggest that the Benford's Law-inspired grid can provide a useful non-uniform codebook for weight quantization, while hardware-level efficiency comparisons against optimized FP4 formats remain a separate question which motivates future work.
>
> 2. Unclear motivations for BenQ: when compared to other data-free methods with non-uniform grids, BenQ accuracy/perplexity are very comparable with NF4, and potentially to FP4 E2M1 or other non-uniform grids. In the scenarios when these methods show better performance than integer quantization, this is usually attributed to the good handling of outliers while preserving near-zero representation. These are well known and studied effects. If BenQ grid doesn't meaningfully improve representation, it is not clear what's the motivation of using yet another log grid, just with different spacing.
>
> We thank you for this thoughtful remark, and we address it along three lines.
>
> First, we agree that BenQ's accuracy and perplexity results are broadly comparable to those of other data-free non-uniform methods such as NF4. We view this comparability as a positioning statement rather than a shortcoming: We position BenQ in our manuscript not as a universal replacement for existing methods, but as a principled, data-free alternative grounded in an explicit analysis of weight distributions, which can be adopted complementarily depending on the deployment context.
>
> Second, while aggregate results may appear similar across methods, a more granular analysis reveals practical scenarios where BenQ or its variants offer meaningful advantages. For instance, on Gemma3-270M, BenQ (GA) achieves a perplexity of 36.455, compared to 40.389 for NF4, a non-trivial gap at this scale. Such differences may be relevant in resource-constrained settings where small models are precisely the deployment target. We believe that a practitioner with awareness of these trade-offs is well-positioned to select the appropriate method.
>
> Third, we respectfully note that the contributions of this paper extend beyond the quantization grid itself. As acknowledged, Benford's Law adherence in Linear layer parameters has been previously reported in the literature, however, to the best of our knowledge, this work is the first to apply the Campanelli test for formal adherence verification in LLMs, and the first to systematically analyze adherence at both the layer-wise and model-wise scopes. This dual-scope analysis (whose importance we have further elaborated in the RQ1 findings of the revised paper, in section 4.1) reveals non-trivial dynamics, such as model families exhibiting layer-wise adherence heterogeneity that is obscured by model-level aggregates, which we argue constitute a meaningful analytical contribution independent of the quantization method itself.

---

> > ### Author Response · Authors · 2026-06-26
> > **Response (part 2/3)**
> >
> > 3. Incomplete characterization of BenQ variant: BenQ(GA) variant defines an individual set of quantization levels for each group. The additional quantization metadata impact compression (as shown in Fig. 4 and Table 14). Does BenQ(GA) has compute overheads? What is the runtime of BenQ(GA) vs BenQ vs NF4 for a sizeable model?
> >
> > We added a subsection (BenQ (GA) computational overhead) in section 4.2 to address this question in the revised paper.
> >
> > As stated, BenQ (GA) defines an individual set of quantization levels for each group, which consequently introduces a computational overhead. To assess it, we conducted an analysis of the throughput (tokens/second) of each method in the perplexity task, varying group-size ($g\in \{64,128\}$) and models.
> >
> > The results (Table 14 and Figure 6 of the revised paper) revealed that the overhead trend is favorable with increasing model size: in the evaluated models, larger architectures exhibit smaller relative slowdowns; and even in Gemma 3-270M, it managed to achieve relatively high throughput (above 8,600 tokens/second). This behavior is consistent with the inference cost becoming increasingly dominated by matrix multiplication and feed-forward computation, which dilutes the relative contribution of per-group grid construction.
> >
> > Regarding the variation of group-size, for all evaluated models the difference in relative overhead between the two configurations ($g\in \{64,128\}$) is generally between 1 and 2 percentage points. This is noteworthy because from 128 to 64 we double the number of groups and consequently we double per-group grid calculations, and, however, the additional cost observed in practice was marginal.
> >
> > Finally, we point out that the reported throughput results provided a conservative assessment, as all measurements were obtained without specialized inference kernels. Dedicated kernels may reduce the observed overhead in practical deployments and, because of that, are left as promising future work.
> >
> > 4. Hyperparameter selection clarification: equations 10 and 11 define the BenQ codebook levels. These levels depend strongly on the choice of epsilon. How was epsilon selected? How does it impact performance?
> >
> > An explanation of how $\epsilon$ is calculated was provided in Section 3 of the revised paper, as well as how important it is for the quantization process.
> >
> > In brief terms, $\epsilon = 10^\omega$, where $\omega = \left\lfloor \log_{10} \left( \frac{Q_{0.999}\!\left(|\mathcal{W}|\right)}{\max|\mathcal{W}|} \right) \right\rfloor$, $Q_{0.999}$ is the quantile at level $0.999$, and $\mathcal{W}$ are the sampled weights from the model (100,000 elements per each nn.Linear tensor). This quantity captures the logarithmic displacement between the bulk of the weight distribution and its absolute peak, yielding a non-positive integer that encodes how many orders of magnitude separate the two.
> >
> > About its importance, we emphasize that early experiments with arbitrarily chosen values of $\epsilon$ yielded severe perplexity degradations, and even a generally well-suited fixed value (\textit{e.g.} $\epsilon=10^{-2}$) failed to generalize across all model families. Therefore, this selection process can be seen as a principled, distribution-aware estimate of $\epsilon$ for BenQ, reducing reliance on arbitrary fixed choices. However, we do not claim or prove optimality in our $\epsilon$ selection process, which remains as an interesting direction for future work.

---

> > > ### Author Response · Authors · 2026-06-26
> > > **Response (part 3/3)**
> > >
> > > 5. Misleading claims about layer-selective quantization: it is repeatedly remarked that Linear layers are quantized with BenQ while LayerNorm and Embeddings are kept in high precision because the former adhere to Benford's Law, while the latter do not. It is not mentioned however that is also the case for the vast majority of quantization algorithms for model deployment (one sentence in Section 4.2 seems to imply otherwise: "In contrast, the models ... framework."). It is not clear then if the increased degradation upon LayerNorm and/or Embeddings quantization (Figure 5) is driven by a failure that is specific to BenQ and lack of adherence to Benford's Law. Similar behaviors are observed under integer quantization.
> > >
> > > First, about the mentioned sentence, we acknowledge that it was not well formulated. Our intention was to explain what methods were implemented by us (BenQ, BenQ GA, RTN and NF4) and which ones used a framework implementation. Therefore, we reformulated the sentence to ``Finally, the models quantized with SOTA methods were either downloaded from HuggingFace or quantized with the AutoAWQ/Transformers framework, which also skips quantization of embeddings and normalization layers by default``.
> > >
> > > Second, we acknowledge that we did not explicitly state that this selective quantization strategy is also the case for many quantization algorithms. To fix this, we added a paragraph in Section 3 (Selective Quantization Strategy subsection) where we explicitly state ``that this policy of excluding \texttt{LayerNorm} and \texttt{Embedding} layers in quantization is not new in literature``. However, we also state that ``this exclusion has traditionally been motivated by practical considerations [...] rather than by any analysis of the underlying weight distributions``. Therefore, we highlight our contribution of grounding this policy in Benford’s Law adherence, and we clarify that the Benford analysis primarily supports the exclusion of normalization layers, while the decision to keep embeddings in higher precision is motivated by vocabulary-interface stability.
> > >
> > > Finally, addressing the concern of whether the increased degradation upon LayerNorm quantization ``is driven by a failure that is specific to BenQ and lack of adherence to Benford's Law``, we agree that degradation upon quantizing normalization layers is not exclusive to BenQ, and, as correctly stated, ``similar behaviors are observed under integer quantization``.
> > >
> > > However, our results reveal an asymmetry that is consistent with the Benford's Law adherence hypothesis. As shown in Table 13 and Figure 5 of the revised paper, quantizing normalization layers with BenQ's log-uniform levels incurs greater perplexity degradations than quantizing them with uniform levels — a gap of approximately 16 perplexity points (uniform: 56.44 vs. log-uniform: 72.41 on DeepSeek-R1-Distill-Qwen-1.5B). This asymmetry is expected under our framework: log-uniform quantization levels are calibrated to the weight distributions of nn.Linear layers, and are therefore particularly mismatched to the near-uniform or spike-dominated distributions of normalization weights. Uniform quantization, lacking this distributional assumption, does not suffer from the same mismatch.
> > >
> > > This contrast thus constitutes additional empirical support for the Benford's Law adherence motivation underlying BenQ's selective quantization strategy, rather than being merely a limitation of its log-uniform levels.
> > >
> > > 6. Additional Comment - use of the term "ablation study" is often imprecise. When used for Bit-width and Group-size (Section 4.2), it refers to a parameter sweep, not to a study where a component is removed / disabled.
> > >
> > > We changed the term ``ablation study`` for ``parameter sweep`` for bit-width and group-size.

---

> > > > ### Comment · Reviewer_u1Z6 · 2026-06-26
> > > > **response to author's answers**
> > > >
> > > > I thank the authors for their response. Following these clarifications and updates to the text, I now hold a more positive view of this paper. I have no further questions.